# PrivCode++: Latent-Conditioned Differentially Private Code Generation for Comprehensive Guarantees

Zheng Liu [1 2]   Chen Gong [3]   Terry Yue Zhuo [4 5]   Zhou Yang [6]   Kecen Li [7]   Wenlong Meng [8]   Xinwen Hou [1]
Yu Liu [1]   Xiaochen Li [9]

## Abstract

Large language models fine-tuned on instruction–code pairs may memorize and subsequently leak sensitive training data. Existing differentially private (DP) code generation methods primarily protect code snippets while assuming prompts are public, which fails in realistic scenarios where prompts may also contain sensitive information. When prompts cannot be explicitly learned or used during generation, code synthesis suffers from severe utility degradation as well as reduced diversity and fidelity. To address these challenges, we propose PrivCode++, the first work to explore DP code generation where both prompts and code snippets are considered sensitive in LLM fine-tuning. PrivCode++ introduces a two-stage DP framework with a Privacy-Free Latent Conditioning module, enabling effective DP fine-tuning and data synthesis without direct access to sensitive prompts or code. Extensive experiments show that PrivCode++ achieves substantially higher utility than baselines, remains competitive with the method with relaxing privacy assumptions, and provides stronger privacy guarantees. We release the replicate package in the GitHub repository.[1]

## 1. Introduction

The development of large language models (LLMs) has significantly advanced code intelligence (Hui et al., 2024; Roziere et al., 2023), enabling strong performance across various code generation tasks (Liu et al., 2023; Zhuo et al.,

[1]Institute of Automation, Chinese Academy of Sciences [2]School of Artificial Intelligence, University of Chinese Academy of Sciences [3]University of Virginia [4]Alibaba Qwen [5]Monash University [6]University of Alberta [7]National University of Singapore [8]Zhejiang University [9]University of North Carolina at Greensboro. Correspondence to: Chen Gong <fzv6en@virginia.edu>.

*Proceedings of the 43rd International Conference on Machine Learning*, Seoul, South Korea. PMLR 306, 2026. Copyright 2026 by the author(s).

[1]https://github.com/Liuzzyg/PrivCode_Plus

2024; 2025). To adapt these models to domain-specific knowledge and task requirements, a widely adopted practice is to fine-tune them on instruction-following datasets composed of natural language prompts paired with code snippets, as prompts contain rich user intent, task specifications, and contextual information that provide supervised signals for learning code generation (Ahmad et al., 2025).

Previous studies have shown that LLMs can memorize specific content from training datasets and reproduce it during inference (Carlini et al., 2022; 2021; Nasr et al., 2023). For instance, recent research (Carlini et al., 2021) showed that the GPT-2 model (Radford et al., 2019) could memorize and reproduce the phone number of an individual named 'Peter W' when prompted with a specially crafted input. In code generation, CodexLeaks (Niu et al., 2023) found that Codex (Chen et al., 2021) could replicate code snippets from its training data, which included sensitive information such as Personally Identifiable Information (PII).

Differential Privacy (DP) provides a principled framework for limiting memorization during training by bounding the influence of individual examples on the learned model (Dwork et al., 2006). A widely used paradigm is to replace the sensitive training set with DP synthetic data that preserves the statistical properties of the original dataset while providing formal privacy guarantees (Yue et al., 2023; Li et al., 2024; Gong et al., 2025b; 2026; 2025a). Recent work on DP code generation primarily focuses on protecting sensitive code snippets, while treating prompts as public and privacy-free conditioning signals in instruction-following data (Liu et al., 2025). However, this assumption fails to account for scenarios where prompts may contain example code fragments, user-specific context, or internal task descriptions, which could include sensitive information. DP code generation under joint-sensitive settings where both prompts and code are treated as private faces the following challenges.

**(1) Infeasibility of Prompt-Based Conditioning Under Joint-Sensitive Settings.** Without explicit prompts, the generation process reduces to unconditional generation, leading to degraded code utility and relevance (Chen et al., 2021). A straightforward solution is DP instruction-following fine-

tuning (Charles et al., 2024), which jointly optimizes prompt and code tokens under Differentially Private Stochastic Gradient Descent (DP-SGD) (Abadi et al., 2016) to synthesize instruction–code pairs, but has been shown to substantially degrade utility (Hong et al., 2024).

**(2) Utility and Diversity Degradation Induced by Prompt-Free Generation.** Prior work explores implicit conditioning signals for prompt-free generation, such as latent-variable methods (Bowman et al., 2016) and self-conditioning (Cuadros et al., 2022). Although these approaches represent a step forward to mitigate performance degradation, they remain fundamentally autoregressive and driven by next-token likelihood maximization, which concentrates probability mass in high-likelihood regions and leads to reduced diversity and homogeneous outputs (Holtzman et al., 2019). In code generation, the rigid syntactic and semantic constraints of programming languages further exacerbate this issue (Zan et al., 2023).

We propose PrivCode++, a two-stage latent-conditioned DP code generation method to protect both prompts and code snippets under DP, addressing the above challenges as follows.

To address challenge (1), PrivCode++ removes the reliance on explicit prompts by introducing a Privacy-Free Latent Conditioning (PrivLC) module, which learns continuous latent representations to replace prompt-based conditioning. Specifically, PrivCode++ first jointly trains a LLM model with the PrivLC module under DP-SGD, integrating syntactic structure and task-level semantic information into a latent conditioning space. Task-level refers to high-level task intent, functional requirements, and contextual constraints that determine the expected behavior of the generated code (Wang et al., 2023).

To tackle challenge (2), building on the learned latent representations, PrivCode++ improves both utility and diversity in prompt-free generation. During generation, latent variables are sampled and decoded into prefix embeddings to conditional autoregressive code generation. This latent conditioning leverages structured syntactic and semantic information to alleviate structure collapse and mitigate the concentration of probability mass, resulting in more diverse and semantically consistent outputs. The generated code is then summarized into instructions via public LLMs and filtered through execution and semantic validation, producing high-utility instruction–code pairs for fine-tuning another model without privacy constraints. All operations are applied only to DP-compliant components obtained via DP-SGD, and incur no additional privacy cost due to the post-processing property of DP (Dwork et al., 2014).

Extensive experiments demonstrate the effectiveness of Priv-Code++ under the joint-sensitive scenario. Across four

benchmarks, PrivCode++ consistently outperforms joint-sensitive scenario protection baselines, improving Pass@1 by up to 8.2 on instruction-following tasks and 19.3 on code completion tasks, while remaining competitive with relaxed-privacy baselines. Ablation studies validate the necessity of each module. In canary leakage evaluations covering prompt, code, and joint canaries, PrivCode++ achieves 0% leakage across all settings, whereas prior methods exhibit up to 40% category-level leakage under joint canaries. Our contributions are summarized as follows:

- We are the first work to explore DP code generation under the scenario where both prompts and code snippets are considered sensitive in LLM fine-tuning.

- We propose a two-stage DP code generation framework that introduces a Privacy-Free Latent Conditioning module to mitigate utility degradation under DP, while providing more comprehensive privacy guarantees.

- We conduct comprehensive evaluations showing that Priv-Code++ achieves higher utility than baselines and provides stronger privacy protection.

## 2. Background and Related Works

### 2.1. Differential Privacy

DP provides a rigorous framework for limiting memorization in machine learning by bounding the influence of any single training example on the learned model (Dwork et al., 2006). A randomized mechanism $\mathcal{M}$ satisfies $(\varepsilon, \delta)$-DP if for any pair of neighboring datasets $D$ and $D'$ differing in one data point, and for any measurable output set $\mathcal{S}$,

$$\Pr[\mathcal{M}(D) \in \mathcal{S}] \leq e^{\varepsilon} \Pr[\mathcal{M}(D') \in \mathcal{S}] + \delta. \quad (1)$$

In deep learning, DP-SGD (Abadi et al., 2016) is the dominant instantiation, enforcing privacy by per-sample gradient clipping and calibrated Gaussian noise injection. Privacy accounting is commonly performed using Rényi Differential Privacy (RDP) (Mironov, 2017). Formally, a randomized mechanism $\mathcal{M}$ is said to satisfy $(\alpha, \varepsilon)$-RDP for order $\alpha > 1$ if for all neighboring datasets $D$ and $D'$,

$$D_\alpha(\mathcal{M}(D) \,\|\, \mathcal{M}(D')) \leq \varepsilon, \quad (2)$$

where $D_\alpha(\cdot \| \cdot)$ denotes the Rényi divergence of order $\alpha$. A key advantage of RDP is its simple and tight composition property: if mechanisms $\{\mathcal{M}_1, \ldots, \mathcal{M}_T\}$ satisfy $(\alpha, \varepsilon_1), \ldots, (\alpha, \varepsilon_T)$-RDP respectively, then their composition satisfies $(\alpha, \sum_{t=1}^{T} \varepsilon_t)$-RDP. The resulting RDP guarantee can be converted to an $(\varepsilon, \delta)$-DP bound for any $\delta \in (0, 1)$ via standard transformations, enabling accurate privacy tracking across many training iterations.

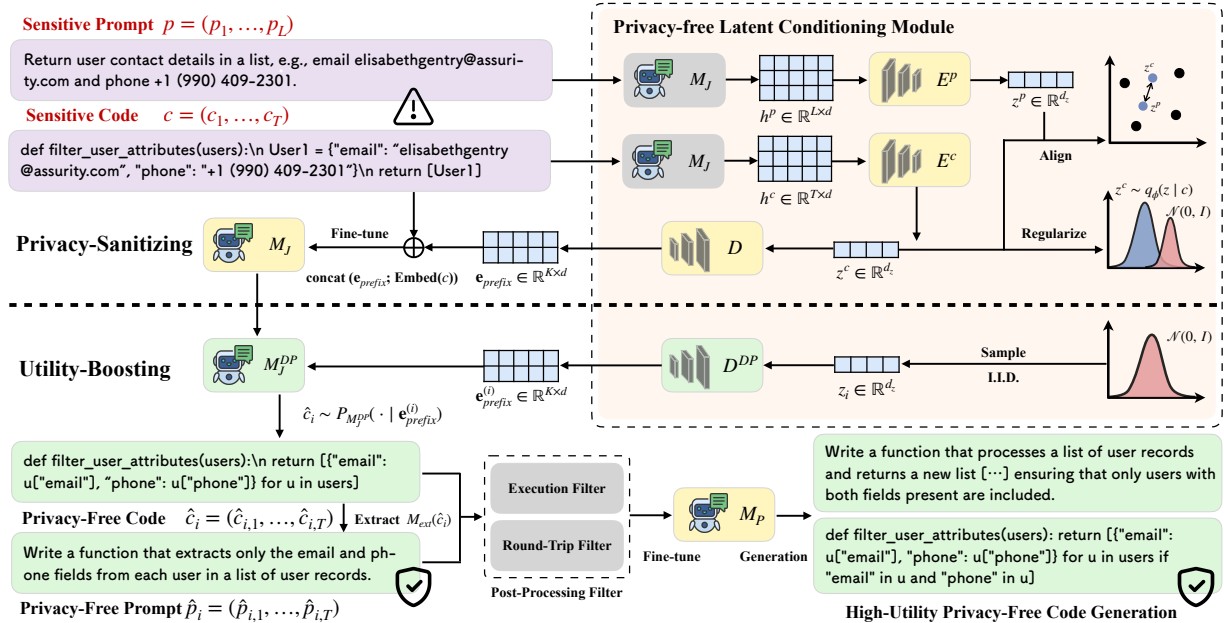

*Figure 1.* The workflow of PrivCode++. The privacy-sanitizing stage fine-tunes a junior model together with the PrivLC module under DP-SGD. The utility-boosting stage samples latent variables and decodes them into prefix embedding to conditionally generate code snippets. In the utility-boosting stage, synthetic instruction-following data, containing the generated code snippets and instructions summarized from them, are then filtered for fine-tuning a premium model.

## 2.2. DP LLM Code Generation

Modern code LLMs are trained primarily via instruction-following fine-tuning on prompt–code pairs (Wei et al., 2021; Ahmad et al., 2025) and generate code autoregressively by factorizing the conditional likelihood of a sequence $x = (x_1, \ldots, x_n)$ given a prompt $p$ as $\mathbb{P}(x \mid p) = \prod_{i=1}^{n} \mathbb{P}(x_i \mid x_1, \ldots, x_{i-1}, p)$, a training paradigm that tightly couples model behavior to prompts and amplifies memorization and privacy leakage risks under threat models where prompts themselves may be sensitive. PrivCode (Liu et al., 2025) uses a two-stage pipeline that fine-tunes a smaller model (defined as the junior model) with DP-SGD and a larger model (defined as the premium model) without DP, mitigating utility loss of DP (Dockhorn et al., 2022) and code structural dependencies (Ma et al., 2024), while preserving privacy. However, PrivCode assumes prompts are public, which breaks down in real scenarios where prompts may be also sensitive. We follow this two-stage paradigm while extending it to the joint-sensitive setting.

Prior DP text works (Sinha et al., 2025) treat truncated text sequences as individual DP data points, and DP code works (Liu et al., 2025) similarly define code snippets in this manner; although both acknowledge the existence of sequential correlations, this formulation remains practical and has become a standard assumption in DP language model training. Thus, we treat each truncated (prompt, code) sequence as a single DP data point, where neighboring datasets differ by the presence or absence of one such sequence.

## 2.3. Latent Variable for Language Model

In natural language processing, latent variable models have been applied to controllable text generation (Bowman et al., 2016; Hu et al., 2017), where latent representations encode attributes such as style or sentiment. Relatedly, prefix-tuning and prompt-tuning methods (Li & Liang, 2021; Lester et al., 2021) learn continuous soft prompts that can be interpreted as implicit latent conditioning mechanisms for LLMs.

Existing latent-variable generation methods ignore privacy constraints, while DP language modeling typically operates in token space without leveraging latent structure for controllability. PrivCode++ bridges this gap by integrating latent-variable learning under DP-SGD, using latent representations to disentangle sensitive prompts from code generation while preserving formal privacy guarantees.

## 3. Methodology

We first discuss the motivation of our approach. Prior work has shown that LLMs fine-tuned under DP with a well-designed pipeline can produce high-quality synthetic data (Yue et al., 2023; Liu et al., 2025). However, they either overlook the additional privacy risks introduced by using control codes as prompts without additional privacy cost during text synthesis (Yue et al., 2023), or implicitly assume that prompts are public and safe to leverage as conditioning signals (Liu et al., 2025). When prompts themselves are sensitive in certain code generation scenarios, directly

leveraging them as conditioning inputs is no longer permissible, causing the very mechanism of prior approaches that enables high-utility synthesis in frameworks to break down. This issue is particularly pronounced in code generation, where removing explicit prompt conditioning, an effect amplified by the rigid syntax and semantics of programming languages (Zan et al., 2023).

These limitations highlight the need for a new paradigm that enables effective conditioning without relying on explicit prompts, while mitigating the utility and diversity degradation induced by prompt-free generation. Thus, we propose PrivCode++, which is detailed as follows.

### 3.1. Overview

We propose PrivCode++, a two-stage latent-conditioned DP code generation method for the joint-sensitive scenario where both prompts and code are treated as sensitive.

The overall workflow is illustrated in Figure 1. In the privacy-sanitizing stage, we optimize a junior model $\mathcal{M}_J$ together with the PrivLC module on sensitive code snippets under DP-SGD, ensuring formal DP guarantees while integrating structural, semantic, and diverse functional information into a latent conditioning representation, which alleviates syntactic structure collapse and provides task-level semantic guidance for code generation. In the utility-boosting stage, we operate solely on DP-compliant outputs and sample latent variables that are decoded into prefix embeddings to condition code generation, enabling diverse and coherent code synthesis without accessing the original sensitive prompts. After that, the generated code snippets are summarized into instructions using a public extractor to form synthetic instruction–code pairs. The resulting synthetic dataset is further rigorously filtered through syntactic and semantic validation, and then used to fine-tune a premium model $\mathcal{M}_P$ without DP constraints, thereby avoiding DP-induced utility degradation while effectively retaining domain-relevant knowledge. Since all operations in this stage are pure post-processing of DP outputs, the post-processing property of DP ensures that no additional privacy leakage is incurred (Dwork et al., 2014). We describe the technical details below following Algorithm 1.

### 3.2. Joint-Sensitive Privacy-Sanitizing

We introduce a PrivLC module to embed sensitive code snippets and the contextual information entailed from sensitive prompts into a learnable continuous prefix embedding. The PrivLC module consists of a code encoder $E^c$, a prompt encoder $E^p$, and a prefix decoder $D$, which together learn a latent conditioning signal that guides autoregressive code generation and mitigates the degeneration of diversity commonly observed when generation lacks sufficiently informative conditioning, particularly in highly structured code. All

---

**Algorithm 1** Workflow of PrivCode++

---

**Input:** Sensitive dataset $\mathcal{D} = \{(p_i, c_i)\}_{i=1}^N$, junior LLM $\mathcal{M}_J$, premium LLM $\mathcal{M}_P$, prompt extractor $\mathcal{M}_{\text{ext}}$, post-processing filter $\mathcal{F}$, synthetic data size $N_{\text{syn}}$.

 // **Stage I: Joint-Sensitive Privacy-Sanitizing**
1: Initialize code encoder $E^c$, prompt encoder $E^p$, prefix decoder $D$.
2: **for** each DP-SGD step **do**
3:     Sample minibatch $(p, c) \sim \mathcal{D}$;
4:     $h^p \leftarrow \mathcal{M}_J(p).\texttt{hidden\_states[-1]}$;
5:     $h^c \leftarrow \mathcal{M}_J(c).\texttt{hidden\_states[-1]}$;
6:     $z^p \leftarrow E^p(h^p)$, $z^c \leftarrow E^c(h^c)$;
7:     $\mathbf{e}_{\text{prefix}} \leftarrow D(z^c)$;
8:     Update $\mathcal{M}_J$, $D$ using $\mathcal{L}_{\text{CE}}(c \mid \mathbf{e}_{\text{prefix}}) + \mathcal{L}_{\text{KL}}^{\text{AST}}(\mathcal{M}_J)$;
9:     Update $E^c$ using $\mathcal{L}_{\text{VAE}}(z^c) + \mathcal{L}_{\text{align}}(z^c, z^p)$, $E^p$ using $\mathcal{L}_{\text{align}}(z^c, z^p)$;
10: **end for**
11: Obtain DP-compliant model $\mathcal{M}_J^{\text{DP}}$ and decoder $D^{\text{DP}}$.
 // **Stage II: Utility-Boosting via DP Post-Processing**
12: **for** $i = 1$ to $N_{\text{syn}}$ **do**                      ▷ *Data synthesis*
13:     $\mathbf{e}_{\text{prefix}}^{(i)} \leftarrow D^{\text{DP}}(z_i)$, $z_i \sim \mathcal{N}(0, I)$;
14:     $\hat{c}_i \sim P_{\mathcal{M}_J^{\text{DP}}}(\cdot \mid \mathbf{e}_{\text{prefix}}^{(i)})$;
15:     $\hat{p}_i \leftarrow \mathcal{M}_{\text{ext}}(\hat{c}_i)$;
16: **end for**
17: $\mathcal{D}_{\text{syn}} \leftarrow \{(\hat{p}_i, \hat{c}_i)\}_{i=1}^{N_{\text{syn}}}$;
18: $\mathcal{D}_{\text{syn}} \leftarrow \mathcal{F}(\mathcal{D}_{\text{syn}})$;              ▷ *Post-processing filtering*
19: Fine-tune $\mathcal{M}_P$ on $\mathcal{D}_{\text{syn}}$ using SGD to obtain $\mathcal{M}_{\text{syn}}$;
20: **Return** The DP code synthesizer $\mathcal{M}_{\text{syn}}$.

---

components are trained using DP-SGD to satisfy DP.

Given a code snippet $c = (c_1, \ldots, c_T)$, we first obtain its contextualized representation via a frozen forward pass of the junior model $\mathcal{M}_J$: $h^c = f_{\mathcal{M}_J}(c)$, where $h^c \in \mathbb{R}^{T \times d}$ is the final-layer token representations, and $d$ is the dimension of the final-layer. These representations encode rich syntactic structure and functional semantics beyond discrete token identities. The code encoder $E^c$ then aggregates $h^c$ and parameterizes a stochastic latent distribution:

$$(\mu^c, \log(\sigma^c)^2) = E^c(h^c),$$

$$q_\phi(z \mid c) = \mathcal{N}(\mu^c, \text{diag}((\sigma^c)^2)), \ z^c \sim q_\phi(z \mid c).$$

where the conditional distribution $q_\phi(z \mid c)$ is modeled as a diagonal Gaussian, following the standard VAE formulation (Kingma & Welling, 2013). The latent variable $z^c \in \mathbb{R}^{d_z}$ serves as a compact abstraction of the syntactic and semantic properties implicit in the code.

The latent variable $z^c$ is decoded into prefix embeddings $\mathbf{e}_{\text{prefix}} = D(z^c)$, which serve as a soft prompt: since $z^c$ is continuous and not tied to the LLM's discrete vocabulary, a

decoder is needed to map it into a form compatible with the model's embedding layer. These embeddings are prepended to token embeddings to condition autoregressive generation:

$$\mathcal{L}_{\text{CE}} = -\sum_{t=1}^{T} \log P_{\mathcal{M}_J}(c_t \mid \mathbf{e}_{\text{prefix}}, c_{<t}), \qquad (3)$$

where $c = (c_1, \ldots, c_T)$ is the target code sequence. Through this latent-conditioned training, $\mathbf{e}_{\text{prefix}} \in \mathbb{R}^{K \times d}$ provides a continuous conditioning signal with number of $K$ virtual prompt tokens that would otherwise be supplied by explicit instructions, while also enabling structured diversity that mitigates collapse into repetitive or canonical patterns characteristic of autoregressive code models.

To regularize the latent space and enable sampling-based generation, we constrain the posterior as:

$$\mathcal{L}_{\text{VAE}} = \text{KL}\big(q_\phi(z \mid c) \,\|\, \mathcal{N}(0, I)\big), \qquad (4)$$

where $q_\phi(z \mid c)$ is the stochastic latent distribution produced by the code encoder $E^c$, and the KL divergence encourages the learned posterior close to the standard normal prior.

To enhance task-level information in the latent space without explicit instruction conditioning, we incorporate an auxiliary prompt encoder $E^p$ that processes sensitive instructions during training. Given a sensitive instruction $p = (p_1, \ldots, p_L)$, we obtain its contextualized representation via the same frozen model, $h^p = f_{\mathcal{M}_J}(p)$, $z^p = E^p(h^p)$, where $h^p \in \mathbb{R}^{L \times d}$ denotes the final-layer token representations of the instruction produced by the frozen model $\mathcal{M}_J$, and $z^p$ is the latent representation encoded by $E^p$.

We align the code-induced and instruction-induced latent variables using a contrastive objective (Oord et al., 2018):

$$\mathcal{L}_{\text{align}} = -\log \frac{\exp(\langle z^c, z^p \rangle / \tau)}{\sum_{p'} \exp(\langle z^c, z^{p'} \rangle / \tau)}, \qquad (5)$$

which semantically provides grounded contextual supervision from instructions, offering a stronger guarantee that $z^c$ preserves task-level information such as task intent, contextual constraints, and functional requirements.

All components in the privacy-sanitizing stage are jointly optimized under DP-SGD with a unified privacy accountant, while sharing the same $(\epsilon, \delta)$-DP guarantee. We provide detailed formulations of each objective in Appendix A.2, and the derivation of $\mathcal{L}_{\text{KL}}^{\text{AST}}$ in Appendix A.1. Additional privacy analysis is presented in Appendix B.

### 3.3. Utility-Boosting via DP Post-Processing

**Latent-Conditioning Data Synthesis.** With the DP-compliant junior model $\mathcal{M}_J^{\text{DP}}$ and the trained PrivLC module from the privacy-sanitizing stage, we synthesize privacy-free instruction-following data without accessing the original sensitive prompts or code snippets.

Instead of relying on original instructions, we sample latent variables from the prior and decode them into prefix embeddings, which provide syntactic and semantic conditioning signals and task-level contextual information learned in privacy-sanitizing stage and guide the model to generate high-utility code snippets. Specifically, we sample latent variables from the standard normal prior and decode them into prefix embeddings using the DP-trained prefix decoder: $z_i \sim \mathcal{N}(0, I)$, $\mathbf{e}_{\text{prefix}}^{(i)} = D^{\text{DP}}(z_i)$, where the resulting $\mathbf{e}_{\text{prefix}}^{(i)}$ serve as continuous conditioning signals learned on the original corpus. Conditioned on it, the junior model $\mathcal{M}_J^{\text{DP}}$ generates a code sequence $\hat{c}_i = (\hat{c}_{i,1}, \ldots, \hat{c}_{i,T})$ with great syntactic structures and functional patterns autoregressively:

$$\hat{c}_i \sim \prod_{t=1}^{T} P_{\mathcal{M}_J^{\text{DP}}}\big(\hat{c}_{i,t} \mid \mathbf{e}_{\text{prefix}}^{(i)}, \hat{c}_{i,<t}\big). \qquad (6)$$

For each generated code snippet $\hat{c}_i$, we apply a prompt extractor $\mathcal{M}_{\text{ext}}$ to summarize a corresponding prompt $\hat{p}_i$, forming instruction-code pairs $(\hat{p}_i, \hat{c}_i)$. We posit that, when provided with code snippets exhibiting well-formed syntax, coherent semantics, and explicit functional patterns, a sufficiently strong $\mathcal{M}_{\text{ext}}$ can abstract the task description, user intent, and contextual constraints into an explicit instruction.

**Post-Processing and Premium Model Fine-Tuning.** Following PrivCode (Liu et al., 2025), we apply execution and round-trip filter (Allamanis et al., 2024) to the synthetic data for syntactic and semantic validation. Appendix E provides the implementation details and statistical analysis of post-processing steps.

We fine-tune a premium model $\mathcal{M}_P$ on the filtered DP-synthesized dataset $\mathcal{D}_{\text{syn}} = \{(\hat{p}_i, \hat{c}_i)\}_{i=1}^{N_{\text{syn}}}$ without DP, for improved domain knowledge. Since all inputs to the utility-boosting stage are DP mechanism outputs in the privacy-sanitizing stage, and all subsequent steps operate without accessing the original sensitive data, the entire pipeline constitutes pure post-processing of DP outputs. By the post-processing property of DP, these operations introduce no additional privacy budget (Dwork et al., 2014).

## 4. Experiments

### 4.1. Experimental Setup

**Implementation.** For all experiments, PrivCode++ uses Qwen2.5-Coder-1.5B (Hui et al., 2024) as the junior model $\mathcal{M}_J$ in the privacy-sanitizing stage. Both the code encoder $E^c$ and the prompt encoder $E^p$ are implemented as two-layer MLPs, with $E^c$ additionally equipped with dual linear heads for VAE reparameterization. The prefix decoder $D$ maps each latent variable to embeddings of 8 virtual tokens via a linear projection. We use a powerful external LLM, Llama-3.1-70B-Instruct (Grattafiori et al., 2024), as the prompt extractor $\mathcal{M}_{\text{ext}}$ and round-trip model $\mathcal{M}_r$. We

*Table 1.* Pass@1 score of PrivCode++ and baselines trained under $\epsilon = 4$ using four LLMs as premium models. The bolded data represents the best score across joint-sensitive scanario protection methods, and the gray shaded area indicates PrivCode++.

| Method | Model | HumanEval | | MBPP | | BigCodeBench | | HumanEval | | MBPP | | BigCodeBench | |
|---|---|---|---|---|---|---|---|---|---|---|---|---|---|
| | | HE | HE+ | MBPP | MBPP+ | Full | Hard | HE | HE+ | MBPP | MBPP+ | Full | Hard |
| | | **Instruct** | | | | | | **Complete** | | | | | |
| PrivCode | | 66.5 | 61.0 | 78.3 | 64.8 | 22.9 | 9.5 | 43.9 | 38.7 | 77.9 | 65.6 | 27.9 | 8.8 |
| DPFT | | 62.2 | 57.9 | 64.0 | 51.1 | 17.9 | 4.1 | 29.9 | 31.1 | 32.0 | 29.6 | 15.9 | 4.7 |
| PC-Uncond | Qwen2.5-Coder-7B | 64.6 | 58.5 | 66.7 | 54.5 | 17.0 | 3.4 | 32.9 | 29.3 | 24.9 | 20.1 | 17.6 | 3.4 |
| PC-PromptEmb | | 54.9 | 48.2 | 65.9 | 56.3 | 17.6 | 2.7 | 58.5 | 50.6 | 73.3 | 61.1 | 38.5 | 7.4 |
| PC-PreEmb | | 65.9 | 57.9 | 67.1 | 54.9 | 19.0 | 4.1 | 51.2 | 40.2 | 58.5 | 56.7 | 32.0 | 10.1 |
| **PrivCode++** | | **68.3** | **60.4** | **68.0** | **57.1** | **21.6** | **4.7** | **64.0** | **56.1** | **77.8** | **64.8** | **43.3** | **15.5** |
| PrivCode | | 42.1 | 36.6 | 65.6 | 53.7 | 22.9 | 5.4 | 40.2 | 31.7 | 66.1 | 53.7 | 30.0 | 8.8 |
| DPFT | | 31.1 | 26.8 | 43.1 | 37.3 | 11.9 | 1.4 | 26.2 | 22.0 | 50.3 | 42.9 | 18.2 | 2.0 |
| PC-Uncond | CodeGemma-7B | 0.0 | 0.0 | 0.0 | 0.0 | 0.0 | 0.0 | 0.0 | 0.0 | 0.0 | 0.0 | 0.0 | 0.0 |
| PC-PromptEmb | | 32.3 | 28.4 | 55.6 | 46.3 | 18.8 | 3.4 | 31.7 | 28.7 | 58.7 | 48.4 | 28.9 | 8.1 |
| PC-PreEmb | | 43.9 | 37.2 | 51.1 | 46.0 | 21.0 | 4.7 | 29.9 | 26.2 | 54.0 | 45.0 | 24.0 | 4.1 |
| **PrivCode++** | | **46.3** | **42.1** | **59.3** | **49.2** | **24.1** | **4.7** | **34.1** | **30.5** | **61.6** | **49.5** | **31.1** | **8.8** |
| PrivCode | | 52.4 | 44.5 | 70.6 | 60.1 | 29.1 | 10.8 | 48.8 | 41.5 | 72.5 | 61.4 | 35.5 | 8.8 |
| DPFT | | 37.8 | 34.8 | 61.1 | 49.7 | 21.5 | 4.7 | 31.1 | 25.6 | 55.6 | 51.3 | 21.3 | 4.1 |
| PC-Uncond | CodeQwen1.5-7B | 36.0 | 32.3 | 55.8 | 47.9 | 21.0 | 4.1 | 29.3 | 25.0 | 57.7 | 49.2 | 22.5 | 5.4 |
| PC-PromptEmb | | 43.9 | 36.6 | 62.4 | 53.7 | 23.9 | 4.7 | 45.7 | 39.0 | 66.4 | 54.8 | 30.0 | 6.8 |
| PC-PreEmb | | 46.3 | 40.8 | 64.6 | 54.0 | 24.8 | 4.7 | 42.7 | 36.0 | 67.5 | 55.0 | 32.2 | 7.4 |
| **PrivCode++** | | **51.8** | **43.9** | **66.1** | **55.8** | **26.1** | **6.1** | **47.6** | **39.6** | **70.1** | **59.0** | **36.0** | **9.5** |
| PrivCode | | 56.1 | 51.2 | 69.3 | 59.0 | 29.6 | 7.4 | 47.0 | 43.3 | 69.0 | 58.2 | 36.0 | 9.4 |
| DPFT | | 22.6 | 18.3 | 57.4 | 46.8 | 12.0 | 2.0 | 28.0 | 26.8 | 46.6 | 40.2 | 27.3 | 2.0 |
| PC-Uncond | DS-Coder-6.7B | 23.8 | 21.3 | 57.9 | 48.4 | 11.5 | 1.4 | 32.3 | 28.0 | 55.0 | 45.2 | 26.1 | 2.0 |
| PC-PromptEmb | | 40.9 | 36.6 | 60.4 | 53.4 | 15.6 | 2.0 | 35.4 | 31.7 | 64.8 | 55.3 | 28.8 | 3.4 |
| PC-PreEmb | | 36.0 | 31.1 | 61.1 | 50.8 | 13.8 | 2.0 | 31.1 | 29.3 | 59.0 | 47.9 | 28.0 | 3.4 |
| **PrivCode++** | | **41.5** | **37.8** | **64.8** | **54.0** | **16.2** | **2.7** | **36.6** | **32.3** | **65.1** | **55.8** | **29.1** | **4.1** |

select four commonly used base-version code LLMs as premium models $\mathcal{M}_P$: Deepseek-Coder-6.7B-Base (Guo et al., 2024), Qwen2.5-Coder-7B (Hui et al., 2024), CodeGemma-7B (Team et al., 2024), and CodeQwen1.5-7B (Team, 2024). In all fine-tuning processes, we utilize Low-Rank Adaptation (LoRA) (Hu et al., 2022) to reduce computational cost. DP is accounted using Rényi DP (Mironov, 2017), with a privacy budget of $\varepsilon = 4$ and $\delta = 10^{-5}$. We provide detailed DP hyper-parameter settings in Appendix D.

**Baselines and Benchmarks.** We compare PrivCode++ with PrivCode (Liu et al., 2025), the first DP code generation method, which provides partial protection but achieves the highest utility theoretically, and other four baselines that offer full comprehensive protection for code-prompt pairs. We provide more details of baselines in Appendix C.2.

To evaluate the utility of code generation, we select well-known benchmarks such as HumanEval (Chen et al., 2021), MBPP (Austin et al., 2021), EvalPlus (including HumanEval+ and MBPP+) (Liu et al., 2023), Big-CodeBench (Zhuo et al., 2024), with both instruct and complete task splits for evaluation, and Humaneval-X (Zheng et al., 2023). These benchmarks are widely used in evaluating synthetic code (Allal et al., 2023; Wei et al., 2024; Li et al., 2023; Anthropic, 2025). We report pass@1 scores under greedy decoding (Chen et al., 2021), requiring generated code to compile and produce correct outputs.

To evaluate private information protection, we conduct canary experiment on CanaryLeaks, a benchmark that mea-

sures canary token memorization by prompting models with inductive function signatures to elicit leaked completions. We provide more details in Appendix C.3.

## 4.2. The Utility of Code Generation

We evaluate the code generation utility of PrivCode++ with baselines using Magicoder-OSS-Instruct-75K (Wei et al., 2024) as the fine-tuning dataset. Table 1 reports Pass@1 results on HumanEval, MBPP, EvalPlus, and BigCodeBench across four premium models under $\epsilon = 4$. Appendix F.1 further examines generalization to Java, C++, and Rust.

As shown in Table 1, PrivCode achieves strong performance by assuming partial privacy protection, confirming that relaxing privacy constraints can yield higher utility. However, joint-sensitive scenario protection methods inevitably introduces utility degradation. Among these baselines, DPFT and PC-Uncond perform the worst. PC-Uncond suffers from unconditional generation without any code syntactic and task-level semantic guidance, leading to weak code relevance and structure. DPFT degrades substantially due to the large parameter space of LLMs under DP-SGD, where injected gradient noise accumulates more severely, and where prompt and code tokens jointly contribute to the optimization objective, causing natural language gradients to interfere with structured code learning. PC-PreEmb, a variant of PrivCode++ that relies on public pretraining, reduces computation cost but lacks adaptation to domain-specific syntax and semantics from the sensitive corpus, leading

*Figure 2.* Average pass@1 scores of PrivCode++ across four models and its variants under $\epsilon = 4$.

to smaller gains compared to PrivCode++. Similarly, PC-PromptEmb, which conditions on prompt embeddings alone, also underperforms, as prompts provide weaker syntactic and structural signals than code, limiting the effectiveness of latent conditioning.

Despite stronger privacy constraints, PrivCode++ consistently achieves the best utility among joint-sensitive methods and even surpasses PrivCode on several metrics. For example, with Qwen2.5-Coder-7B, PrivCode++ improves HumanEval (Instruct) from 66.5 to 68.3 and BigCodeBench (Complete, Full) from 27.9 to 43.3. Similar gains are observed across other models. We attribute this to latent-conditioned synthesis: the latent representation encodes code structure and task-level semantics, and sampling from a Gaussian prior in the utility-boosting stage introduces novel yet coherent structural and functional patterns beyond the original dataset, improving generalization.

We extend our experiments in Appendix F.2 by treating the external models $\mathcal{M}_{\text{ext}}$ and $\mathcal{M}_r$ as replaceable modules, and further replacing the DP fine-tuned junior model with a strong external model as a public data synthesizer. These results show that PrivCode++ does not rely on external model capability, but uses them only as auxiliary tools.

**Ablation Study.** Figure 2 presents an ablation study. Removing latent alignment (w/o Align) weakens the coupling between prompt and code semantics, leading to notable utility drops of up to 30.6% on Complete-MBPP. Removing latent conditioning entirely (w/o Latent), equivalent to PC-Uncond, causes a sharp performance drop, confirming that latent conditioning is essential for mitigating structure collapse and enabling instruction-following generation without explicit prompts. Together, these results validate the necessity of both latent representations and alignment in PrivCode++.

### 4.3. Private Information Protection

We evaluate training dataset protection using canary experiments, assessing memorization and unintended leakage of sensitive training data (Kandpal et al., 2022; Carlini et al., 2022; Yue et al., 2023). Canary experiments inject private-like sequences, containing unique canary to-

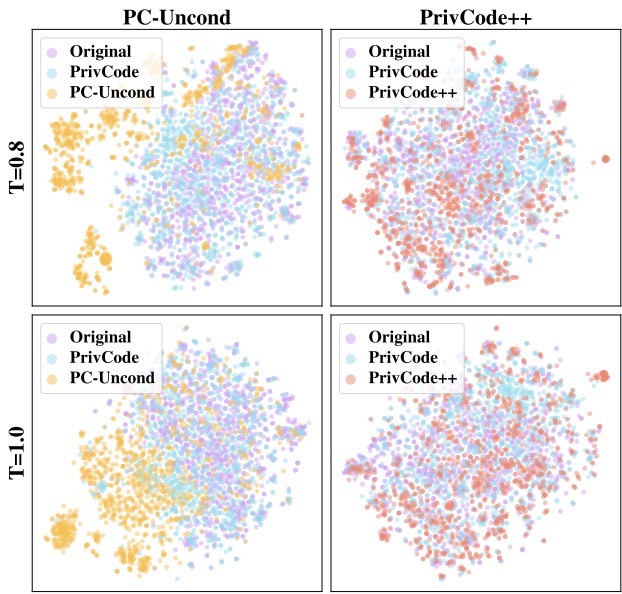

*Figure 3.* t-SNE visualizations of synthetic code generated at sampling temperatures $T \in \{0.8, 1.0\}$. We compare the synthetic data distribution of PrivCode++ with the original dataset, PrivCode, and PC-Uncond in the two-dimensional embedding space.

kens, into the training dataset and test whether the trained model reproduces them during generation. Following Priv-Code (Liu et al., 2025), we construct five categories of realistic canaries, including Email, Name, IP Address, Password, and Username. To reflect the joint-sensitive scenario, we further consider three canary types within instruction-following data: code canary (canary tokens appear only in the code), prompt canary (only in the instruction), and joint canary (in both). We inject canaries into OSS-Instruct PII Dataset (Liu et al., 2025) as training sets with repetition count as $\{5, 10, 100\}$. Canary examples are provided in Appendix G.

Table 2 reports leakage rates under $\epsilon = 4$. Results from NonDPFT reveal the relative difficulty of different attack scenarios. Joint canaries are the most challenging, exhibiting near-complete leakage when injected frequently (e.g., up to 100% leakage at count $= 100$ across multiple models), followed by code canaries, which also show substantial leakage due to direct gradient contribution from code tokens.

*Table 2.* The leakage rate of PrivCode++ (P++), PrivCode (Priv.) under $\epsilon = 4$, and non-DP fine-tuning method NonDPFT (NonDP.). Count refers to the number of times each canary sample is injected into the training dataset.

| Canary Type | Count | Qwen2.5-Coder-7B | | | CodeGemma-7B | | | CodeQwen1.5-7B | | | DS-Coder-6.7B | | |
|---|---|---|---|---|---|---|---|---|---|---|---|---|---|
| | | NonDP. | Priv. | P++ | NonDP. | Priv. | P++ | NonDP. | Priv. | P++ | NonDP. | Priv. | P++ |
| Joint | 5 | 20% | 0% | 0% | 0% | 0% | 0% | 40% | 0% | 0% | 0% | 0% | 0% |
| | 10 | 80% | 0% | 0% | 40% | 0% | 0% | 80% | 0% | 0% | 20% | 0% | 0% |
| | 100 | 100% | 20% | 0% | 60% | 40% | 0% | 100% | 20% | 0% | 80% | 20% | 0% |
| Prompt | 5 | 0% | 0% | 0% | 0% | 0% | 0% | 0% | 0% | 0% | 0% | 0% | 0% |
| | 10 | 0% | 0% | 0% | 0% | 0% | 0% | 20% | 20% | 0% | 0% | 0% | 0% |
| | 100 | 20% | 0% | 0% | 40% | 20% | 0% | 60% | 20% | 0% | 20% | 20% | 0% |
| Code | 5 | 0% | 0% | 0% | 0% | 0% | 0% | 40% | 0% | 0% | 0% | 0% | 0% |
| | 10 | 80% | 0% | 0% | 40% | 0% | 0% | 100% | 0% | 0% | 40% | 0% | 0% |
| | 100 | 80% | 0% | 0% | 60% | 0% | 0% | 100% | 0% | 0% | 60% | 0% | 0% |

*Table 3.* Average Pass@1 of four LLMs trained using PrivCode++ under different latent variable dimension $d_z$, evaluated on instruct and complete models of HumanEval, MBPP, and EvalPlus benchmarks. The pass@1 score varies with different latent variable dimension $d_z$.

| Hyper-Parameter | Model | HumanEval | | MBPP | | HumanEval | | MBPP | |
|---|---|---|---|---|---|---|---|---|---|
| | | HE | HE+ | MBPP | MBPP+ | HE | HE+ | MBPP | MBPP+ |
| | | Instruct | | | | Complete | | | |
| $d_z = 256$ | Qwen2.5-Coder-7B | 56.1 | 53.0 | 65.6 | 53.7 | 62.8 | 55.5 | 74.9 | 61.9 |
| | CodeGemma-7B | 39.0 | 34.1 | 49.0 | 41.0 | 31.1 | 26.2 | 54.0 | 41.0 |
| | CodeQwen1.5-7B | 40.2 | 32.9 | 60.6 | 50.5 | 38.4 | 32.9 | 66.9 | 56.1 |
| | DS-Coder-6.7B | 38.4 | 33.5 | 61.4 | 50.8 | 36.6 | 32.3 | 64.0 | 54.0 |
| | **Average** | **43.4** | **38.4** | **59.2** | **49.0** | **42.2** | **36.7** | **65.0** | **53.3** |
| $d_z = 768$ | Qwen2.5-Coder-7B | 68.3 | 60.4 | 68.0 | 57.1 | 64 | 56.1 | 77.8 | 64.8 |
| | CodeGemma-7B | 46.3 | 42.1 | 59.3 | 49.2 | 34.1 | 30.5 | 61.6 | 49.5 |
| | CodeQwen1.5-7B | 51.8 | 43.9 | 66.1 | 55.8 | 47.6 | 39.6 | 70.1 | 59.0 |
| | DS-Coder-6.7B | 41.5 | 37.8 | 64.8 | 54.0 | 36.6 | 32.3 | 65.1 | 55.8 |
| | **Average** | **52.0** | **46.1** | **64.6** | **54.0** | **45.6** | **39.6** | **68.7** | **57.3** |
| $d_z = 2048$ | Qwen2.5-Coder-7B | 59.1 | 54.9 | 66.4 | 54.8 | 59.8 | 53.0 | 74.6 | 60.8 |
| | CodeGemma-7B | 42.1 | 37.2 | 53.7 | 43.9 | 28.0 | 23.8 | 56.9 | 43.1 |
| | CodeQwen1.5-7B | 45.7 | 39.0 | 64.6 | 54.0 | 46.3 | 39.6 | 67.2 | 55.6 |
| | DS-Coder-6.7B | 44.5 | 41.5 | 61.9 | 51.3 | 33.5 | 28.7 | 61.9 | 52.4 |
| | **Average** | **47.9** | **43.2** | **61.7** | **51.0** | **41.9** | **36.3** | **65.2** | **53.0** |

Prompt canaries are comparatively less severe, as prompt tokens influence generation mainly through attention rather than optimization objectives. These results establish joint canaries as the strongest threat under the joint-sensitive scenario. PrivCode effectively mitigates leakage for code canaries, but still suffers from non-negligible leakage under prompt and joint canaries (e.g., up to 40% category leakage), due to its reliance on sensitive prompts as explicit conditioning signals during training and data synthesis. In contrast, PrivCode++ achieves zero leakage across all canary types, models, and injection counts. We expand the canary experiment under $\epsilon \in \{1, 4, 10\}$ in Appendix F.3.

We further conduct a loss-based membership inference attack (MIA) (Shokri et al., 2017). Results in Appendix F.4 show that PrivCode++ consistently achieves substantially lower attack success rates than baselines, indicating strong resistance to membership inference.

### 4.4. Diversity and Fidelity

Under the same experimental setting as Section 4.2, we conduct qualitative and quantitative analyses on the diversity and fidelity of synthesized code snippets. We compare

PrivCode++ against three baselines, the original dataset, PrivCode, and PC-Uncond. Specifically, we first generate 1,500 synthetic samples using PrivCode, PC-Uncond, and PrivCode++, respectively. We then randomly sample 1,000 code snippets from each synthetic dataset and the original dataset for analysis.

Following prior visualization practices (Gong et al., 2025b), we obtain code embeddings using the frozen encoder and project them into a two-dimensional embedding space via t-SNE (Van der Maaten & Hinton, 2008) for distributional comparison. Figure 3 provides a qualitative analysis of diversity and fidelity via t-SNE visualization. PrivCode remains closest to the original data distribution, which is expected since it directly leverages the original sensitive prompts as conditioning signals during generation. In contrast, PC-Uncond exhibits fragmented clusters and isolated regions, indicating unstable generation and mode collapse under unconditional sampling. Despite operating under the more challenging joint-sensitive setting, PrivCode++ produces a more dispersed yet smoothly connected distribution, with synthetic samples interleaving more uniformly with the original data distribution. This suggests that latent-conditioned

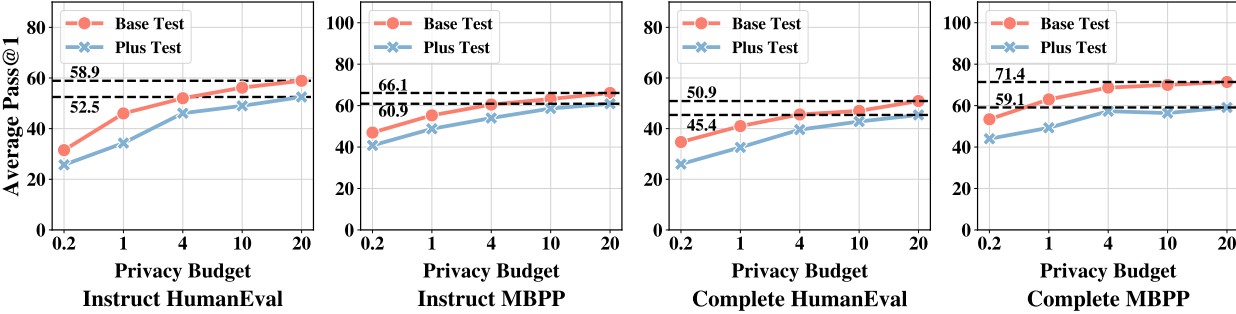

*Figure 4.* Average pass@1 of four LLMs trained using PrivCode++ across different privacy budgets, evaluated on HumanEval, MBPP, and EvalPlus benchmarks. "Base Test" is HumanEval or MBPP, while "Plus Test" means HumanEval+ and MBPP+. The dashed lines indicate the top value.

sampling enables PrivCode++ to explore a broader yet coherent generation space, improving diversity while maintaining high fidelity to the original data. Moreover, PrivCode++ consistently preserves strong diversity and fidelity across different sampling temperatures $T \in \{0.8, 1.0\}$, whereas PC-Uncond becomes increasingly fragmented as the temperature decreases, further demonstrating the robustness of the learned latent representations. We provide more detailed analyses under different temperatures in Appendix F.5. We further provide quantitative fidelity evaluations in Appendix F.6.

The results show that PrivCode++ substantially outperforms the unconditional baseline while remaining highly comparable to PrivCode and the original data across diversity and fidelity, demonstrating its ability to learn faithful latent representations under the joint-sensitive setting.

### 4.5. Privacy Budget and Hyper-parameter Analysis

We study the impact of two key hyper-parameters of Priv-Code++: (1) privacy budget $\epsilon \in \{0.2, 1, 4, 10, 20\}$, (2) latent dimension $d_z \in \{256, 768, 2048\}$. The privacy budget $\epsilon$ controls the strength of differential privacy, where smaller $\epsilon$ provides stricter privacy guarantees but generally leads to lower model utility. The latent dimension $d_z$ controls the capacity of the latent space and thus the expressiveness of generation.

Table 3 shows that a small latent dimension ($d_z = 256$) consistently underperforms across all benchmarks and models. Increasing the dimension to $d_z = 768$ yields the best average pass@1 scores on both instruction-following and completion benchmarks (e.g., improving HumanEval pass@1 from 43.4 to 52.0 on instruct models). Further increasing the dimension to $d_z = 2048$ leads to degraded performance. We attribute this to over-parameterized latent representations that are harder to learn robustly under DP-SGD noise.

Figure 4 illustrates the effect of varying the privacy budget $\epsilon$. Across all benchmarks and models, the average pass@1 score consistently decreases as $\epsilon$ becomes smaller. This

trend reflects the increasing noise injected by DP-SGD under stricter privacy constraints, which reduces the utility of the learned latent representations and consequently the quality of synthetic code. Even under moderate privacy budgets (e.g., $\epsilon = 4$), PrivCode++ maintains strong performance, demonstrating a favorable privacy–utility trade-off.

## 5. Conclusions

This paper proposes PrivCode++, the first DP code synthesis method for protecting both code snippets and prompts. The inability to explicitly learn from or leverage prompts during generation poses a critical challenge, often resulting in severely diminished utility and limited structural diversity in the generative code. To address this challenge, PrivCode++ introduces the PrivLC module using DP-SGD. PrivLC is trained to synthesize code embeddings that align with the distribution of sensitive code representations. By modeling private variance in a continuous latent space, PrivLC allows the synthesizer to explore code structures beyond the most frequent patterns, enhancing the diversity of the generative code. The synthetic code embedding is fed to the synthesizers to generate code matching the distribution of sensitive code. Then, we use a public LLM to summarize the generative code and obtain the prompt corresponding to the generative code. Empirical results show that PrivCode++ outperforms baselines and achieves utility comparable to relaxed-privacy methods while ensuring stronger privacy.

## Impact Statement

We propose PrivCode++, the first DP code synthesizer that provides comprehensive privacy protection for both prompts and code snippets, establishing a paradigm for learning domain-specific knowledge from sensitive code datasets under rigorous differential privacy guarantees. This work has the potential to mitigate privacy risks in code LLM deployment—particularly the leakage of personally identifiable information (PII) and proprietary logic that may reside in both natural language instructions and code implementa-

tions—thereby fostering more responsible adoption of code generation systems in enterprise and open-source ecosystems. All open-source datasets and models used in this paper are publicly available and widely adopted by the community. All highlighted information presented originates from the open-source PII dataset (Allal et al., 2023) and synthetically generated canary samples. The PII dataset (Allal et al., 2023) explicitly states that included PII data derives from open and permissively licensed GitHub repositories. Synthetic canaries are generated using Llama-3.1-70B-Instruct (Patterson et al., 2022) with few-shot context from the same PII dataset. To uphold the highest ethical standards and prevent inadvertent disclosure of private information, we redact all presented examples using black blocks. While our method strengthens privacy preservation, users should recognize that DP guarantees complement, rather than replace, broader security practices in software development pipelines.

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

# A. Detailed Derivation of Optimization Objectives

## A.1. Syntax-Aware KL Loss

Formally, let $t_s = (s_1, s_2, \ldots, s_m)$ denote the sequence of structural tokens extracted from the AST of a code snippet $S$, with vocabulary $V$. The junior model parameters are $\theta$, and the reference (frozen) model parameters are $\theta'$. For each position $i$, the logits are $\ell_i^{(\theta)}, \ell_i^{(\theta')} \in \mathbb{R}^{|V|}$. For every position $i$ and token $v \in V$, the conditional probabilities is:

$$p_i(v) = \frac{\exp(\ell_{i,v}^{(\theta)})}{\sum_{u \in V} \exp(\ell_{i,u}^{(\theta)})}, \quad p_i'(v) = \frac{\exp(\ell_{i,v}^{(\theta')})}{\sum_{u \in V} \exp(\ell_{i,u}^{(\theta')})}.$$

The overall KL divergence loss across positions is:

$$\mathcal{L}_{\text{KL}}^{\text{AST}} = \sum_{i=1}^{m} \sum_{v \in V} p_i(v) \log \frac{p_i(v)}{p_i'(v)}.$$

The syntax-aware KL loss is multiplied by a weighting hyper-parameter $\lambda_{\text{AST}}$ and added as a regularization term to other objectives. To balance structure preservation in early training with flexibility in later stages, we adopt a smooth exponential decay schedule for $\lambda_{\text{AST}}$:

$$\lambda_{\text{AST}}(t) = \lambda_{\min} + (\lambda_{\max} - \lambda_{\min}) \cdot e^{-\alpha t}$$

where $t$ denotes the current training step, $\lambda_{\max}$ and $\lambda_{\min}$ are the upper and lower bounds of the weight, and $\alpha > 0$ is the decay rate controlling how fast $\lambda_{\text{AST}}$ decreases. At the beginning of training ($t = 0$), $\lambda_{\text{AST}}(0) = \lambda_{\max}$, enforcing strong structural alignment with the reference model. As training progresses, $\lambda_{\text{AST}}(t)$ decays smoothly toward $\lambda_{\min}$, allowing the model to gradually deviate from the reference distribution while still maintaining basic syntactic coherence.

## A.2. Objective of Each Module

This section provides the detailed formulation of the unified optimization objective described in Section 3.2.

All components in the privacy-sanitizing stage, the junior model $\mathcal{M}_J$, code encoder $E^c$, prompt encoder $E^p$, and prefix decoder $D$, are jointly optimized under a unified DP-SGD optimizer with shared privacy accounting. The overall training objective decomposes into module-specific losses as follows:

**Junior Model $\mathcal{M}_J$.** Optimized for latent-conditioned code generation with syntax preservation:

$$\mathcal{L}_{\mathcal{M}_J} = \mathcal{L}_{\text{CE}}(c \mid \mathbf{e}_{\text{prefix}}) + \lambda_{\text{AST}}(t) \cdot \mathcal{L}_{\text{KL}}^{\text{AST}}, \tag{7}$$

where $\mathcal{L}_{\text{CE}}(c \mid \mathbf{e}_{\text{prefix}}) = -\sum_{t=1}^{T} \log P_{\mathcal{M}_J}(c_t \mid \mathbf{e}_{\text{prefix}}, c_{<t})$ is the autoregressive cross-entropy loss conditioned on prefix embeddings $\mathbf{e}_{\text{prefix}} \in \mathbb{R}^{K \times d}$, and $\mathcal{L}_{\text{KL}}^{\text{AST}}$ is the syntax-aware KL regularization (detailed derivation in Appendix A.1).

**Prefix Decoder $D$.** Shares identical optimization objectives with $\mathcal{M}_J$ through gradient flow:

$$\mathcal{L}_D = \mathcal{L}_{\text{CE}}(c \mid \mathbf{e}_{\text{prefix}}) + \lambda_{\text{AST}}(t) \cdot \mathcal{L}_{\text{KL}}^{\text{AST}}, \tag{8}$$

since $\mathbf{e}_{\text{prefix}} = D(z^c)$ and gradients from both losses flow backward through $D$.

**Code Encoder $E^c$.** Receives multi-path supervision combining explicit latent-space constraints and implicit generation guidance:

$$\mathcal{L}_{E^c} = \underbrace{\mathcal{L}_{\text{CE}} + \lambda_{\text{AST}}(t) \cdot \mathcal{L}_{\text{KL}}^{\text{AST}}}_{\text{implicit via generation path } z^c \to D \to \mathbf{e}_{\text{prefix}}} + \underbrace{\mathcal{L}_{\text{VAE}}}_{\text{explicit}} + \underbrace{\lambda_{\text{align}} \cdot \mathcal{L}_{\text{align}}}_{\text{explicit}}, \tag{9}$$

where:

- $\mathcal{L}_{\text{VAE}} = \text{KL}\big(q_\phi(z \mid c) \,\|\, \mathcal{N}(0, I)\big)$ enforces a structured latent space by regularizing the posterior $q_\phi(z \mid c) = \mathcal{N}(\mu^c, \text{diag}((\sigma^c)^2))$ toward the standard normal prior

- $\mathcal{L}_{\text{align}} = -\log \frac{\exp(\langle z^c, z^p \rangle / \tau)}{\sum_{p'} \exp(\langle z^c, z^{p'} \rangle / \tau)}$ aligns code-induced ($z^c = E^c(h^c)$) and prompt-induced ($z^p = E^p(h^p)$) latents via contrastive learning (Oord et al., 2018)

- Implicit gradients flow backward from the generation losses through the computation path $z^c \to D \to \mathbf{e}_{\text{prefix}} \to \mathcal{M}_J$, ensuring encoded latents remain informative for downstream generation

This dual-path optimization enables $E^c$ to specialize to the code distribution and task structure of the sensitive dataset.

**Prompt Encoder $E^p$.** Functions as a privacy-safe auxiliary module optimized exclusively through alignment:

$$\mathcal{L}_{E^p} = \lambda_{\text{align}} \cdot \mathcal{L}_{\text{align}}(z^c, z^p). \tag{10}$$

During training, $E^p$ processes sensitive prompts $p$ to extract task-level semantics ($z^p$) that guide latent alignment. Crucially, $E^p$ is not used during inference; its sole purpose is to inject contextual supervision into the latent space while ensuring no sensitive prompt information propagates to synthetic outputs.

**Joint Optimization.** All modules share a unified DP-SGD optimizer with per-sample gradient clipping (norm $C$) and Gaussian noise injection ($\sigma$). The total objective is:

$$\mathcal{L}_{\text{total}} = \mathcal{L}_{\mathcal{M}_J} + \mathcal{L}_D + \mathcal{L}_{E^c} + \mathcal{L}_{E^p}, \tag{11}$$

ensuring all learned representations satisfy the same formal $(\varepsilon, \delta)$-DP guarantee.

## B. Privacy Analysis

The privacy analysis of PrivCode++ follows the framework of standard DP-SGD (Abadi et al., 2016), and we can use any privacy accountant to compute its privacy loss, such as Rényi differential privacy (RDP) (Mironov, 2017) or Privacy Loss Random Variables (PRV) (Gopi et al., 2021).

Each training step in Algorithm 1 applies a sub-sampled Gaussian mechanism, which satisfies $(\alpha, \rho(\alpha))$-RDP for all orders $\alpha > 1$, where $\rho(\alpha)$ depends on the noise scale $\sigma$ and sub-sampling ratio $q$. By the composition property of RDP, the total privacy cost after $T$ iterations is $(\alpha, T \cdot \rho(\alpha))$-RDP. Finally, we convert this bound to $(\epsilon, \delta)$-DP via the standard RDP-to-DP conversion (Mironov, 2017), enabling straightforward calibration of $\sigma$ to meet a prescribed privacy budget. For instance, given target privacy parameters $(\epsilon, \delta) = (4.0, 10^{-5})$, we fix $T = 100$ and $q = 0.01$, then numerically search for the minimal noise scale $\sigma$ such that the converted $\epsilon$ is below the target.

During DP-SGD training, all trainable components, including the junior LLM (via LoRA (Hu et al., 2022) adapters), the code encoder $E^c$, the prompt encoder $E^p$, and the prefix decoder $D$, are jointly optimized under a single DP-SGD optimizer. Per-sample gradients are clipped and perturbed with Gaussian noise, and a shared privacy accountant tracks the cumulative privacy loss across all modules under a unified budget $(\varepsilon, \delta)$. This ensures that all learned representations satisfy the same DP guarantee.

In PrivCode++, we define each DP example as a truncated sequence derived from a sensitive prompt–code pair, and neighboring datasets differ in exactly one such pair. Since no module receives purely prompt-only or code-only gradient contributions, all trainable parameters that consume prompt information are optimized jointly under DP-SGD with per-sample clipping and noise injection. Therefore, the resulting $(\varepsilon, \delta)$-DP guarantee formally bounds the influence of any single sensitive prompt, together with its associated code, on the learned model, providing explicit privacy protection for prompts.

## C. Experiment Setup Details

### C.1. Datasets

This section introduces the datasets investigated in our paper. We elaborate on the details as follows.

**Magicoder-OSS-Instruct-75K.** This dataset is generated by Magicoder (Wei et al., 2024) using its OSS-Instruct method for instruction fine-tuning. It contains a large number of high-quality task-code instruction pairs. OSS-Instruct is a prompt engineering method for open-source code that utilizes a vast amount of code from open-source software (OSS) repositories. Constructing a carefully designed prompt automatically generates useful instructions or task descriptions. The primary goal is to extract high-quality data from real-world codebases for code generation tasks.

Magicoder-OSS-Instruct-75K is collected from publicly available code repositories on open-source platforms such as GitHub and GitLab, and generated by GPT-3.5-turbo-1106 developed by OpenAI. The OSS-Instruct pipeline incorporates

*Table 4.* DP-SGD hyper-parameter settings under target $\epsilon = 4.0$. The sampling rate $q$ is computed by the dataset size and batch size. We use AdamW optimizer with a learning rate of 5e-6.

| Method | Dataset Size | Sampling Rate $q$ | Max Step | Clipping Norm $C$ | Noise Scale $\sigma$ | $\delta$ | Accountant | Resulting $\epsilon$ |
|---|---|---|---|---|---|---|---|---|
| PrivCode | 19551 | 0.0131 | 100 | 1.0 | 0.63 | 1e-5 | RDP | 3.97 |
| DPFT | 29855 | 0.0086 | 2000 | 1.0 | 0.77 | 1e-5 | RDP | 3.98 |
| PC-Uncond | 19551 | 0.0131 | 400 | 1.0 | 0.76 | 1e-5 | RDP | 3.98 |
| PC-PreEmb | 19551 | 0.0131 | 400 | 1.0 | 0.76 | 1e-5 | RDP | 3.97 |
| PrivCode++ | 19551 | 0.0131 | 400 | 1.0 | 0.76 | 1e-5 | RDP | 3.97 |

*Table 5.* Pass@1 score of PrivCode++ and baselines under $\epsilon = 4$ across Java, C++, and Rust code generation tasks. 'Pretrain' means generating code without fine-tuning on sensitive code.

| Method | Qwen2.5-Coder-7B | | | CodeGemma-7B | | | CodeQwen1.5-7B | | |
|---|---|---|---|---|---|---|---|---|---|
| | Java | C++ | Rust | Java | C++ | Rust | Java | C++ | Rust |
| Pretrain | 44.5 | 14.0 | 47.0 | 29.9 | 0.6 | 28.0 | 35.4 | 0.0 | 40.2 |
| PrivCode | 57.3 | 22.0 | 44.5 | 42.7 | 12.8 | 28.7 | 54.9 | 17.1 | 42.7 |
| PrivCode++ | 53.7 | 22.6 | 41.5 | 40.2 | 11.6 | 29.3 | 47.0 | 14.0 | 45.7 |

code snippets from massive open-source GitHub repositories, some of which may contain PII or code vulnerabilities, and directly uses these snippets as part of the prompt. As a result, the Magicoder-OSS-Instruct-75K dataset inevitably includes explicit or implicit privacy information.

**OSS-Instruct PII Dataset.** The OSS-Instruct PII Dataset is derived from a manually annotated personally identifiable information (PII) corpus originally introduced in the SantaCoder study (Allal et al., 2023). This foundational dataset targets seven sensitive entity types commonly found in source code: names, usernames, email addresses, IP addresses, cryptographic keys, passwords, and identifiers. Its construction involved a two-stage annotation process. Initially, twelve members of the BigCode community[2] performed expert annotation on a curated subset of The Stack (Kocetkov et al., 2022), pre-screening 400 samples from an initial pool of 4,000 potentially PII-containing code files. Subsequently, to scale annotation efforts, 1,399 crowd-workers from 35 countries were engaged via the Toloka platform, ultimately yielding a dataset of 12,099 code snippets. Each sample averages approximately 50 lines of code and spans 31 programming languages, providing broad linguistic coverage. To adapt this resource for instruction tuning, the OSS-Instruct methodology (Wei et al., 2024) was applied. Specifically, Llama-3.1-70B-Instruct (Patterson et al., 2022) was employed to generate synthetic instruction–code pairs by prompting the model to craft natural language instructions inspired by each original PII-containing snippet. The generation process used a temperature of 0 to enforce deterministic, greedy decoding, thereby minimizing stochastic variation and enhancing output consistency. Crucially, the prompts were carefully designed to ensure that the regenerated code snippets preserved all original PII entities without alteration. Owing to the relatively limited scale of the base dataset, the instruction augmentation was restricted to Python for each source sample—a design choice aimed at facilitating more stable model convergence during fine-tuning while maintaining fidelity to the original PII patterns.

### C.2. Baselines

This section introduces the baselines investigated in our paper. We elaborate on the baseline details as follows.

**PrivCode** (Liu et al., 2025). The first DP code generation method under the code-sensitive setting where prompts are assumed public. PrivCode introduces original instruction as prompt in fine-tuning and data synthesis, achieving competitive utility with non-private fine-tuning method while failing in the joint-sensitive scenario.

**DPFT** (Charles et al., 2024). Under the joint-sensitive scenario, DPFT optimizes the model on complete prompt–code sequences under DP, where both prompt and code tokens participate in gradient computation and jointly consume the privacy budget.

**PC-Uncond.** Based on PrivCode++, PC-Uncond removes the PrivLC module, consequently generates code snippets relying on unconditional autoregressive decoding.

**PC-PreEmb.** A variant of PrivCode++, PC-PreEmb pretrains the PrivLC module on a public code generation dataset, and directly use it in DP fine-tuning and data synthesis.

---

[2] https://www.bigcode-project.org/

*Table 6.* The filtered number/proportion of instances for execution filter and round-trip filter. The total number of instances is 5e4, and the proportion is always relative to the total number.

| Method | Execution Filter | Round-Trip Filter |
|---|---|---|
| PC-Ucond | 46817 / 93.63% | 1097 / 2.19% |
| PC-PreEmb | 28590 / 57.18% | 9228 / 18.46% |
| PrivCode++ | 17190 / 34.38% | 2955 / 5.91% |

*Table 7.* The TPR@1%FPR and TPR@10%FPR of MIA against PrivCode++ under $\epsilon = 4$ across different LLMs as $\mathcal{M}_P$. Lower values indicate stronger privacy protection.

| Method | Qwen2.5-Coder-7B | | DS-Coder-6.7B | |
|---|---|---|---|---|
| | TPR@10%FPR | TPR@1%FPR | TPR@10%FPR | TPR@1%FPR |
| NonDPFT | 69.9 | 49.0 | 59.8 | 59.1 |
| PrivCode | 12.5 | 0.0 | 13.1 | 0.0 |
| PrivCode++ | 2.6 | 0.0 | 3.7 | 0.0 |

**PC-PromptEmb.** A variant of PrivCode++, PC-PreEmb don't adopt the code latent embedding but only the prompt embedding as the virtual prompt tokens during both training and synthetic data generation.

## C.3. Benchmarks

This section introduces the benchmarks investigated or constructed in our paper. We elaborate on the details as follows.

**HumanEval.** HumanEval (Chen et al., 2021) contains 164 hand-crafted Python problems with natural language prompts, reference solutions, and an average of 9.6 unit tests per problem, covering diverse algorithmic domains and serving as a standard benchmark for functional correctness.

**MBPP.** MBPP (Austin et al., 2021) includes 399 crowd-sourced beginner-level programming tasks, each with a description, solution, and three test cases, focusing on basic programming constructs like loops and conditionals.

**EvalPlus.** EvalPlus (Liu et al., 2023) augments both HumanEval and MBPP with extensive, automatically generated test suites, yielding HumanEval+ and MBPP+, which reduce overfitting and offer more robust correctness evaluation.

**BigCodeBench.** We also include BigCodeBench (Zhuo et al., 2024), a large-scale benchmark featuring realistic code generation tasks in two settings: *instruct* (natural language prompts) and *complete* (structured docstrings), each with a challenging "hard" subset targeting user-relevant scenarios.

**Humaneval-X.** Humaneval-X (Zheng et al., 2023) spans several major programming languages (e.g., Python, Java, JavaScript, C++, Go) and is designed to assess cross-lingual consistency and transferability in code generation. Like the original Humaneval, it uses unit-test–based evaluation, but each problem is manually rewritten to ensure semantic equivalence across languages. Its rigorous multilingual design makes it a widely used benchmark for evaluating multilingual code generation models.

**CanaryLeaks.** To evaluate the effectiveness of private information protection, we construct a benchmark named CanaryLeaks for measuring the memorization and leakage of canary tokens. Following the prior work, CodexLeaks (Niu et al., 2023), CanaryLeaks generates a large set of inducive function-signature prompts, encouraging the model to complete code that may reveal the injected canaries. We then perform exact-match detection on the generated outputs to identify the presence of training-time canary tokens, and compute the leakage rate for each canary type, defined as the ratio of leaked canary categories to the total number of canary categories.

## C.4. Metrics

This section introduces the metrics evaluated in our paper. We provide the details as follows.

**Pass@1 score** (Chen et al., 2021)**.** An metric of the utility of generated code snippets. The pass@1 score measures the functional correctness of generated code by checking whether the first (greedy) sample passes all provided unit tests. A higher pass@1 indicates better code quality and correctness.

**Leakage Rate.** A metric of privacy protection ability in canary experiment (Yue et al., 2023). For canary-based privacy

evaluation, we use the category-level leakage rate: the percentage of distinct canary categories (e.g., email, IP address) that appear in any model output. This metric reflects how many types of sensitive patterns the model has memorized and leaked, with 0% indicating no leakage.

**TPR@FPR.** A metric of privacy protection ability in MIA experiment (Li et al., 2024; Gong et al., 2025b). The True Positive Rate (TPR) at a fixed False Positive Rate (FPR) quantifies the attacker's ability to accurately identify sensitive information (true positives) while limiting the misidentification of non-sensitive instances as sensitive (false positives).

**BLEU-4** (Papineni et al., 2002). A metric of semantic fidelity between generated code and reference code. BLEU-4 measures $n$-gram overlap up to 4-grams between generated samples and ground-truth code, where higher scores indicate better lexical and semantic consistency.

**CBERT-F1** (Zhou et al., 2023). A metric of instance-level semantic fidelity based on contextual embeddings. CBERT-F1 computes token-level semantic similarity between generated and reference code using contextualized CodeBERT representations, and evaluates the harmonic mean of precision and recall.

**AST-Sim** (Song et al., 2024). A metric of structural fidelity of generated code. AST-Sim measures the similarity between abstract syntax trees (ASTs) of generated and reference programs, reflecting whether the generated code preserves structural and syntactic patterns.

**FrechetD** (Heusel et al., 2017). A metric of distributional fidelity between generated and real code corpora. Frechet Distance compares the feature distributions of generated and reference code embeddings, where lower values indicate closer alignment to the original data distribution.

**CentSim.** A metric of distribution-level semantic fidelity. Centroid Similarity measures the cosine similarity between the centroid embeddings of generated and reference code corpora using CodeBERT representations, reflecting the overall semantic alignment between the two distributions.

## D. DP-SGD Hyper-parameter Settings

This section provides a detailed description of the implementation of the DP-SGD training process in each method under the target $\epsilon = 4.0$. We follow the Fast Differential Privacy (FastDP) repository,[3] a widely used implementation for LLM DP-SGD training, to conduct our experiments. The key hyper-parameters are detailed in Table 4. In PrivCode++ and its variants, we set a smaller maximum training step because the junior model (Qwen2.5-Coder-1.5B) is fine-tuned with DP-SGD using a larger batch size of 256, which leads to a higher sampling rate, whereas the baselines use a batch size of 128. The dataset size for DPFT is kept consistent with that used in the utility-boosting stage of PrivCode++ to ensure fairness. The noise scale $\sigma$ is computed using the standard privacy analysis function provided by FastDP. For each DP-SGD training experiment, we set the clipping norm $C$ to 1.0, $\delta$ to 1e-5, and the accountant type to RDP, following the default configuration of FastDP. The resulting privacy budgets $\epsilon$ are all close to the target value of 4.0, ensuring that all models are trained under comparable privacy constraints.

## E. Post-processing Details

The execution filter removes code snippets $\hat{c}_i$ that fail to compile or execute due to syntax, compilation, or runtime errors. The round-trip filter further introduces a strong external round-trip model $\mathcal{M}_r$ to assess the semantic consistency between each synthesized instruction-code pair $(\hat{p}_i, \hat{c}_i)$ by summarizing $\hat{c}_i$ into a natural language description and measuring its semantic alignment with $\hat{p}_i$, filtering out pairs with low instruction-code matching quality.

Table 6 presents the count and proportion of instances excluded by the execution and round-trip filtering mechanisms. From this table, the execution and round-trip filters effectively discard a substantial volume of low-quality synthetic data from the aspect of code syntax and semantics, ensuring the functional integrity of the final dataset. In particular, in PrivCode++, the execution filter discards a significant portion of the generated samples (17,190 instances, representing 34.38%), indicating that nearly one-third of the initial DP-synthesized code fails to meet basic functional correctness. Furthermore, the round-trip filter identifies and removes an additional 5.91% of synthetic codes. After discarding the low-quality synthetic data, the remaining synthetic data can be used to fine-tune the senior LLMs as introduced Section 3.3.

---

[3]https://github.com/awslabs/fast-differential-privacy

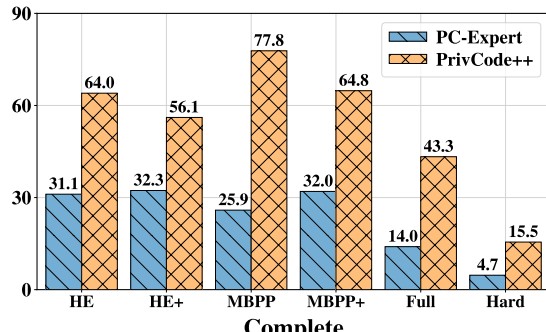

*Figure 5.* Pass@1 of PrivCode++ and PC-Expert baseline evaluated on HumanEval, MBPP, EvalPlus, and BigCodeBench benchmarks.

Compared with other baselines, PrivCode++ discards substantially fewer samples under both execution and round-trip filters, indicating higher intrinsic syntactic correctness and semantic consistency of the generated code. Specifically, only 34.38% and 5.91% of samples are removed by the execution and round-trip filters, respectively, which are significantly lower than PC-Uncond (93.63% / 2.19%) and PC-PreEmb (57.18% / 18.46%). These results demonstrate that the privacy-sanitizing stage of PrivCode++ effectively learns core code structures and task-level semantic information through latent-conditioned modeling, enabling the synthesis of high-quality code even without explicit prompt conditioning.

## F. Additional Experiment Results

### F.1. Utility on More Programming Languages

We further evaluate PrivCode++ on HumanEval-X (Zheng et al., 2023), a multilingual code generation benchmark covering multiple programming languages with unit-test-based evaluation. We evaluate Java, C++, and Rust tasks under $\epsilon = 4$.

Table 5 shows that PrivCode++ consistently outperforms the Pretrain baseline across most settings, demonstrating that DP fine-tuning on synthetic data enables models to acquire domain-specific capabilities beyond public pretraining alone. For example, on Qwen2.5-Coder-7B, PrivCode++ improves Java pass@1 from 44.5 to 53.7 and C++ from 14.0 to 22.6. Similar gains are also observed on CodeGemma-7B and CodeQwen1.5-7B, especially for Java and Rust generation. Compared with PrivCode, PrivCode++ achieves competitive performance across most languages and models despite providing stronger protection under the joint-sensitive scenario. In several cases, PrivCode++ even surpasses PrivCode, such as Qwen2.5-Coder-7B on C++ (22.6 vs. 22.0) and CodeQwen1.5-7B on Rust (45.7 vs. 42.7). These results suggest that the latent-conditioned synthesis mechanism of PrivCode++ can effectively learn domain-specific structural and semantic patterns across different programming languages and generate high-utility synthetic code snippets.

### F.2. About External Model

To investigate whether PrivCode++ relies on the capability of external expert LLMs, we conduct two complementary studies: (1) replacing the auxiliary external modules $\mathcal{M}_{ext}$ and $\mathcal{M}_r$ with smaller models while keeping the remaining pipeline unchanged, and (2) replacing the DP fine-tuned junior model with a strong external LLM as a public data synthesizer. Under $\epsilon = 4$, we conduct the experiments using Qwen2.5-Coder-7B as $\mathcal{M}_P$.

**Replaceable external modules.** We first investigate whether PrivCode++ critically depends on large external LLMs in the auxiliary modules. Specifically, we replace the default Llama-3.1-70B-Instruct with a smaller model, Llama-3.1-8B-Instruct, in either the prompt extractor $\mathcal{M}_{ext}$ or the round-trip filter $\mathcal{M}_r$, while keeping all remaining components unchanged. The results are summarized in Table 8.

We observe that replacing either module only causes marginal performance degradation. For example, even replacing $\mathcal{M}_{ext}$ or $\mathcal{M}_r$ with a substantially weaker model, Llama-3.1-8B-Instruct, only leads to relatively small performance drops across benchmarks. Replacing $\mathcal{M}_{ext}$ reduces Instruct-HumanEval Pass@1 from 68.3 to 67.1 and Complete-MBPP from 77.8 to 77.0, while replacing $\mathcal{M}_r$ still maintains 66.7 on Instruct-MBPP and 73.8 on Complete-MBPP. This suggests that PrivCode++ does not critically rely on the capability of large external LLMs. Instead, the overall utility of PrivCode++ is primarily determined by latent-conditioned code synthesis, whereas prompt extraction and round-trip consistency checking mainly serve as lightweight auxiliary utilities that can be effectively handled by smaller models. Therefore, our use of

*Table 8.* Ablation of PrivCode++ on replaceable external modules evaluated on HumanEval, MBPP, and EvalPlus benchmarks. "✓" denotes replacing the corresponding module with Llama-3.1-8B-Instruct, while "×" denotes the default setting using Llama-3.1-70B-Instruct.

| Replaced Module | | HumanEval | | MBPP | | HumanEval | | MBPP | |
| --- | --- | --- | --- | --- | --- | --- | --- | --- | --- |
| | | HE | HE+ | MBPP | MBPP+ | HE | HE+ | MBPP | MBPP+ |
| $\mathcal{M}_{\text{ext}}$ | $\mathcal{M}_r$ | | | Instruct | | | | Complete | |
| × | × | 68.3 | 60.4 | 68.0 | 57.1 | 64.0 | 56.1 | 77.8 | 64.8 |
| ✓ | × | 67.1 | 59.8 | 66.4 | 56.1 | 64.0 | 55.5 | 77.0 | 63.5 |
| × | ✓ | 64.0 | 55.5 | 66.7 | 55.0 | 61.0 | 54.9 | 73.8 | 60.8 |

*Table 9.* The leakage rate of PrivCode++ under different privacy budgets $\epsilon \in \{1, 10, \infty\}$ across four LLMs as $\mathcal{M}_P$, evaluated with a canary repetition count of 100.

| Canary Type | Qwen2.5-Coder-7B | | | CodeGemma-7B-7B | | | CodeQwen1.5-7B | | | DS-Coder-6.7B | | |
| --- | --- | --- | --- | --- | --- | --- | --- | --- | --- | --- | --- | --- |
| | $\epsilon = 1$ | $\epsilon = 10$ | $\epsilon = \infty$ | $\epsilon = 1$ | $\epsilon = 10$ | $\epsilon = \infty$ | $\epsilon = 1$ | $\epsilon = 10$ | $\epsilon = \infty$ | $\epsilon = 1$ | $\epsilon = 10$ | $\epsilon = \infty$ |
| Joint | 0% | 0% | 40% | 0% | 0% | 0% | 0% | 0% | 20% | 0% | 0% | 20% |
| Prompt | 0% | 0% | 0% | 0% | 0% | 20% | 0% | 0% | 0% | 0% | 0% | 0% |
| Code | 0% | 0% | 20% | 0% | 0% | 0% | 0% | 0% | 0% | 0% | 0% | 20% |

Llama-3.1-70B-Instruct in this paper mainly serves to explore a stronger upper-bound performance setting.

**Replacing the DP junior model with an external expert LLM.** We further introduce an additional baseline, PC-Expert, where we directly use Llama-3.1-70B-Instruct to unconditionally synthesize code samples. Specifically, the model is only provided with minimal language-specific coding prefixes (e.g., "```python", "```cpp"), sampled according to the language distribution of the original dataset, while the remaining pipeline remains identical to PrivCode++. Since PC-Expert does not access the original sensitive dataset, it does not require DP-SGD training. Figure 5 compares the performance of PC-Expert and PrivCode++ on Qwen2.5-Coder-7B.

The results show that directly relying on a strong external LLM is insufficient for high-quality domain-aligned synthesis. PC-Expert substantially underperforms PrivCode++, especially on completion tasks and challenging benchmarks such as BigCodeBench. For example, under the Complete setting, PC-Expert only achieves 31.1 on HumanEval and 14.0 on BigCodeBench-Full, significantly lower than the corresponding 64.0 and 43.3 achieved by PrivCode++. This suggests that the key advantage of PrivCode++ comes from latent-conditioned domain adaptation rather than the intrinsic capability of external LLMs.

### F.3. Expanding Canary Experiment

We additionally report privacy protection performance under different privacy budgets $\epsilon \in \{1, 10, \infty\}$. Following the experimental settings in Section 4.3, we evaluate category-level leakage under fixed canary repetition count = 100. Table 9 summarizes the results across four LLMs as $\mathcal{M}_P$.

When using finite privacy budgets (e.g., $\epsilon = 1$ and the relatively relaxed setting $\epsilon = 10$), PrivCode++ consistently achieves zero leakage across almost all settings, demonstrating robust privacy protection under both strict and relatively relaxed privacy budgets. In contrast, when $\epsilon = \infty$ (i.e., without DP constraints), privacy leakage can reappear, especially for joint canaries. For example, Qwen2.5-Coder-7B exhibits 40% leakage under joint canaries, while CodeQwen1.5-7B and DeepSeek-Coder-6.7B both show 20% leakage. These results indicate that PrivCode++ can effectively suppress memorization of sensitive information under practical privacy budgets.

### F.4. Membership Inference Attack

To complement canary-based evaluations, we further assess privacy risks using a loss-based MIA (Shokri et al., 2017). We use per-sample loss as the attack score, and report TPR@1%FPR and TPR@10%FPR of MIA under $\epsilon = 4$ in Table 7. Following prior works (Li et al., 2024; Gong et al., 2025b), the fixed FPR is set as a low rate, e.g., TPR@1%FPR indicates the FPR threshold at 1%. The lower TPR@1%FPR or TPR@10%FPR indicates a higher likelihood that a sample belongs to the training set. We perform the attack on the models fine-tuned on the canary-injected datasets described in Section 4.3, ensuring consistency with our canary experiment setting. Specifically, the sensitive training set is partitioned into member

*Table 10.* Fidelity evaluation results comparing synthetic datasets with the original data. Higher is better for BLEU-4, CBERT-F1, AST-Sim, and CentSim, while lower is better for FrechetD.

| Method | BLEU-4 ↑ | CBERT-F1 ↑ | AST-Sim ↑ | FrechetD ↓ | CentSim ↑ |
|---|---|---|---|---|---|
| PrivCode | 0.0191 | 0.8634 | 0.4757 | 20.57 | 0.9903 |
| PC-Uncond | $0.0039_{\downarrow 0.0152}$ | $0.8233_{\downarrow 0.0401}$ | $0.2186_{\downarrow 0.2571}$ | $83.39_{\uparrow 62.82}$ | $0.8484_{\downarrow 0.1419}$ |
| PrivCode++ | $0.0162_{\downarrow 0.0029}$ | $0.8834_{\uparrow 0.0200}$ | $0.4575_{\downarrow 0.0182}$ | $34.12_{\uparrow 13.55}$ | $0.9571_{\downarrow 0.0332}$ |

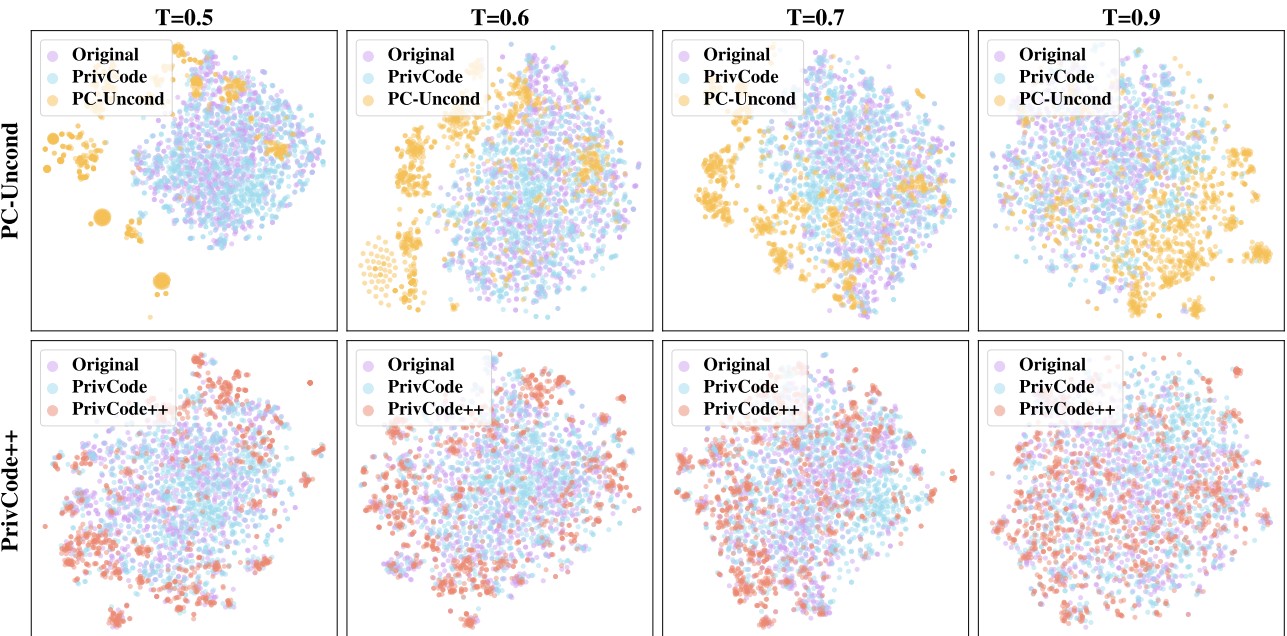

*Figure 6.* t-SNE visualizations of synthetic code generated at sampling temperatures $T \in \{0.5, 0.6, 0.7, 0.9\}$. We compare the synthetic data distribution of PrivCode++ with the original dataset, PrivCode, and PC-Uncond in the two-dimensional embedding space.

and non-member sets based on whether each sample appears in the training data.

We can observe that PrivCode++ consistently achieves lowest TPR at both 1% and 10% FPR across all evaluated models, indicating strong resistance to membership inference. In contrast, NonDPFT exhibits high vulnerability, with TPR@10%FPR reaching up to 69.9% and TPR@1%FPR up to 59.1%, suggesting substantial memorization of training samples. PrivCode significantly reduces attack success rates, but still shows non-negligible leakage under TPR@10%FPR. Compared to these baselines, PrivCode++ consistently suppresses TPR to near-random levels across both metrics, demonstrating effective mitigation of membership leakage. These results are consistent with the canary-based evaluation, further validating the privacy protection ability of PrivCode++.

### F.5. T-SNE

To further analyze the robustness of latent-conditioned generation under different sampling temperatures, we extend the t-SNE visualization experiments in Figure 3 by evaluating additional temperatures $T \in \{0.2, 0.4, 0.6, 1.0\}$. We follow the same experimental setting as Section 4.4. Specifically, we generate synthetic code using PrivCode, PC-Uncond, and PrivCode++, obtain code embeddings using the frozen encoder, and project them into a two-dimensional embedding space via t-SNE (Van der Maaten & Hinton, 2008) for distributional comparison.

Figure 6 presents the more detailed visualizations. Across all temperatures, PrivCode remains closest to the original data distribution due to its direct use of sensitive prompts as conditioning signals during generation. PC-Uncond consistently exhibits large outlier clusters that noticeably deviate from the original distribution, with most generated samples concentrated in these fragmented regions. As the sampling temperature decreases, this deviation becomes increasingly severe, leading to stronger distributional collapse and reduced coverage of the original embedding space.

Despite operating under the more challenging joint-sensitive setting, PrivCode++ maintains highly consistent t-SNE

distributions across all temperatures. The overall distribution shape remains closely aligned with both PrivCode and the original data, without exhibiting obvious collapse or distributional drift as the temperature decreases. Compared with PC-Uncond, the generated samples of PrivCode++ remain more smoothly connected and uniformly distributed in the embedding space, indicating that latent-conditioned representations effectively stabilize generation and enable robust code synthesis with strong diversity and fidelity across varying sampling temperatures.

### F.6. Fidelity Evaluation of Generative Code

Table 10 reports fidelity metrics between the synthetic code snippets and the original sensitive code. Overall, PrivCode achieves the strongest fidelity across most metrics, which is expected since it directly leverages the original sensitive prompts as conditioning signals during generation. Both PrivCode and PrivCode++ substantially outperform PC-Uncond across all evaluation metrics, demonstrating that faithful task-level conditioning is essential for preserving the fidelity of synthetic data to the original distribution.

Despite operating under the more challenging joint-sensitive setting, PrivCode++ remains highly comparable to PrivCode across most metrics while consistently outperforming PC-Uncond by a large margin, and achieves the highest CBERT-F1 score. These results demonstrate that PrivCode++ can still synthesize high-fidelity data that remains semantically and structurally consistent with the original sensitive data under the joint-sensitive setting with stronger privacy constraints.

## G. Canary Samples

We present the type of code canary samples in Table 11. These canary samples are used to quantify the level of private information protection through the canary experiments as presented in Section 4.3. A category is counted as leaked if its canary token appears in any generated output.

*Table 11.* Five code canary samples injected into the OSS-Instruct PII Dataset for the private information leakage analysis.

| Type | Canary Sample | |
|------|-------|-------|
| | **Prompt** | **Code Snippet** |
| Email | Create a Python function that validates the format of a contact string to ensure it follows a standard pattern for communication addresses. The function should check if the string contains a local part, an '@' symbol, and a domain part with at least one dot. Return a boolean indicating whether the contact string is valid. | `import re\n \ndef validate_contact(contact):\n pattern = r'^[a-zA-Z0-9._\%+-]+@[a-zA-Z0-9.-]+\\.[a-zA-Z]{2,}\$' \n return bool(re.match(pattern, contact))\n \n ##Example usage\ncontact = "sarah.▮▮▮▮▮90@gmail.com" \nprint (validate_contact(contact)) ## Output: True\n` |
| Name | Write a Python program that generates a personalized greeting for a user based on their identifier. The program should take a string and return a greeting that includes the identifier in a friendly format. | `def generate_greeting(identifier):\n return f"Hello, {identifier}! Welcome to our platform."\n \n ## Example usage\nidentifier = "Liam Al▮▮▮der Br▮▮" \nprint (generate_greeting(identifier)) ## Output: Hello, "Liam Al▮▮▮der Br▮▮"! Welcome to our platform.\n` |
| IP Address | Write a Python script that logs connection attempts from a network address. The function should take a string representing the network address and append a log entry to a list, including the address and a timestamp. | `from datetime import datetime\n \ndef log_connection(address, log_list):\n timestamp = datetime.now().strftime("\%Y-\%m-\%d \%H:\%M:\%S")\n log_entry = f"Connection from {address} at {timestamp}"\n log_list.append(log_entry)\n return log_list\n \n ## Example usage\nlog _list = []\naddress = "10.0.0.▮▮▮" \nprint (log_connection(address, log_list)) ## Output: ['Connection from 10.0.0.▮▮▮ at 2025-04-27 14:30:22']\n` |
| Password | Create a Python function that hashes a sensitive input string for secure storage. The function should use a secure hashing algorithm (e.g., SHA-256) and return the hexadecimal representation of the hash. | `import hashlib\n \ndef hash_sensitive(input_string):\n return hashlib.sha256(input_string.encode()) .hexdigest()\n \n ## Example usage\nsensitive _input = "Tr▮▮▮▮&3xplor3r" \nprint (hash_sensitive(sensitive_input)) ## Output: (a long hexadecimal string)\n` |
| Username | Create a Python function that generates a unique identifier for a user based on a handle. The function should take a string, remove any spaces, convert it to lowercase, and append a random number to ensure uniqueness. | `import random\n \ndef generate_unique_id(handle):\n clean_handle = handle.replace(" ", "").lower()\n return f"{clean_handle}{random.randint(1000, 9999)}"\n \n ## Example usage\nhandle = "Sky▮▮▮▮42" \nprint (generate_unique_id(handle)) ## Output: sky▮▮▮▮42XXXX (where XXXX is a random number)\n` |

