# OpenReview forum: "PrivCode++: Latent-Conditioned Differentially Private Code Generation for Comprehensive Guarantees"
_ICML.cc/2026/Conference — ICML 2026 regular_

### Official Review · Reviewer_tTVa · 2026-02-26

**Soundness:** 3
**Presentation:** 3
**Significance:** 2
**Originality:** 3
**Overall Recommendation:** 2
**Confidence:** 4

**Summary:**

The paper proposes a novel private mechanism to fine-tune the Code Language Model while protecting the privacy of the prompt and the generated code. The proposed mechanism operates in two phases: (1) Privately Learn a VAE model to generate a prefix that captures the prompt and generated code in the training data, such that the prefix can be used as a latent context and the Code LM can generate code from the prefix, (2) Use the VAE to generate synthetic data to optimize the Code LM. The paper conducts extensive experiments to show the performance of the proposed method.

**Compliance With Llm Reviewing Policy:**

Affirmed.

**Ethical Review Concerns:**

The paper use fabricated citation

**Ethical Review Flag:**

Flag this paper for an ethics review.

**Ethics Expertise Needed:**

["Responsible Research Practice (e.g., IRB, documentation, research ethics)"]

**Final Justification:**

I thank the author for their answers to my question. However, the privacy definition answer is not convincing to me due to the fact that the literature is well-acknowledged as having this problem, and there are several solutions to resolve this, such as UeDP [1], Client-levelDP[2]. Therefore, I suggest that the author clearly define at what level of privacy they provide and steer their mechanism to that specific level.

For these reasons, I maintain my score.

[1] Lai, Phung, et al. "User-Entity Differential Privacy in Learning Natural Language Models." 2022 IEEE International Conference on Big Data (Big Data). IEEE, 2022.
[2] H. McMahan, D. Ramage, K. Talwar, and L. Zhang, “Learning differentially private recurrent language models,” ICLR, 2017.

**Key Questions For Authors:**

- What if a PII (name, emails) appears in multiple data points in a training dataset, then the privacy guarantee of DPSGD is broken. How do you scale your mechanism to this setup?
- In your experiments, the performance does not scale with the dimension size of the embedding. Why is it? Why are over-parameterized latent representations harder to learn robustly under DP-SGD noise?
- How many runs do you run the experiments? Since DP-SGD is a stochastic mechanism, it's better to provide the results across multiple runs.
- In Figure 2 and the PC-Uncond baseline, if you remove the latent embedding, how can the model generate the code? How valid is this baseline since you do not feed the prompt or any information into the LLM?
- The alignment step does not contribute much to the performance since the performance drop is small, according to Figure 2. So I have a question: if you don't adopt the latent embedding in the code but only the prompt embedding, would your mechanism still achieve good performance at a similar privacy leakage rate?

**Limitations:**

Please see the Weakness.

**Strengths And Weaknesses:**

Strengths:
- The technique proposed by the paper is interesting, sound, and novel. By privately learning a VAE to capture the prompt-code distribution. The experiments are well-designed.
- The paper addresses an emerging problem of privacy in the LLM training process.
- The paper is well-written and easy to follow.

Weaknesses:
- The paper lack of theoretical analysis for the privacy guarantee. Since the paper considers DP (a theoretical privacy framework), the authors should derive the theoretical guarantee.
- The privacy mechanism considered in the proposed method (DP-SGD) cannot address the privacy of the prompts and code content. Specifically, DP-SGD provides membership privacy, i.e., the attacker cannot infer with high confidence whether a data point is in the training dataset. Thus, the privacy mechanism does not directly protect the prompt and the code. As a result, the paper lacks comprehension of why the mechanism works.
- The experimental results do not highlight the advantages of using the proposed method compared with the existing baselines. For instance, in Table 1, PrivCode achieves better performance in many tasks. Reducing the impact of the proposed method.
- The paper used a fabricated citation. For example, Liu et al. (2024), which was cited as published in NeurIPS 2024, was actually published in NeurIPS 2023.

---

> ### Author Rebuttal · Authors · 2026-03-31
>
> Q1: Privacy guarantee analysis.
>
> A1: All components (Junior model + PrivLC) are jointly optimized under a single DP-SGD optimizer, so the privacy guarantee follows standard DP-SGD theory. We use RDP accounting and convert to $(\epsilon, \delta)$-DP; data point and neighboring definitions are in Sec. 2.2, with full analysis in App. B.
>
> Q2: Membership privacy.
>
> A2: Our goal is data point-level protection under standard DP, not content-level privacy. As defined in Sec. 2.2 and App. B, each truncated (prompt, code) pair is one DP data point.
>
> Following prior work [a], we use leakage rate (recall) as an MIA metric, where leaking a reproduced canary token implies its sample is inferred as a member, corresponding to a successful inference. We add loss-based MIA, conducted on models trained with canary-injected data: we split the sensitive dataset into member/non-member sets (based on training inclusion), inject canaries into the member set, and use per-sample loss as the attack score. We report TPR@1%FPR / TPR@10%FPR (%):
>
> |Method|Qwen2.5-Coder-7B|DS-Coder-6.7B|
> |-|-|-|
> |NonDPFT|69.9/49.0|59.8/59.1|
> |PrivCode|12.5/0.0|13.1/0.0|
> |PrivCode++|2.6/0.0|3.7/0.0|
>
> TPR@10%FPR (<10%) for PrivCode++ is due to the inherently higher loss of canary samples, as they are generated in a few-shot manner from sensitive data and thus exhibit slight distribution shift. Overall, results show strong privacy protection of Privcode++.
>
> [a]https://aclanthology.org/2023.acl-long.74.pdf
>
> Q3: Advantages over baselines.
>
> A3: PrivCode and PrivCode++ assume different privacy settings (public prompts vs. joint-sensitive). Under this stricter setting, PrivCode++ outperforms all joint-sensitive baselines, remains competitive with PrivCode, and achieves 0% leakage (while PrivCode leaks under prompt/joint canaries).
>
> Q4: Incorrect citation.
>
> A4: Thank you for pointing this out; this is an incorrect year annotation rather than a fabricated citation, and we will correct it.
>
> Q5: PII appears repeatedly.
>
> A5: We agree DP cannot hide population-level statistics when sensitive patterns repeat extensively (e.g., gender attributes in sensitive image datasets [b]), a known limitation often addressed via preprocessing. In our setting, sensitive occurrences are limited; empirically, even with 100 repeated canaries, PrivCode++ achieves 0% leakage (Tab. 2), demonstrating strong robustness and scalability.
>
>
> [b]https://www.usenix.org/system/files/usenixsecurity24-li-kecen.pdf
>
> Q6: Embedding dimensions.
>
> A6: Prior work [b,c] shows that under the same privacy budget, larger models require more noise and are more prone to training instability. Consistent with this, larger latent dimensions make noise dominate learning, leading to a trade-off between expressiveness and robustness; in our setting, $d_z = 768$ achieves a good balance.
>
> [c]https://arxiv.org/pdf/2210.09929
>
> Q7: Multiple runs.
>
> A7: We report single-turn runs in the paper. We agree multiple runs are important under DP-SGD, and add 5-seed results (mean ± std) provided in this [link](https://anonymous.4open.science/r/PrivCode_Plus-02FE/images/multiruns.png). PrivCode++ remains stable with consistent conclusions.
>
> Q8: Validity of PC-Uncond.
>
> A8: PC-Uncond performs autoregressive decoding from an empty token. Due to the lack of standard baselines that avoid prompt usage and leakage, since we are the first to study this joint-sensitive setting, it serves as a naive yet meaningful lower bound without using sensitive inputs. We further introduce PC-PreEmb, which learns latent prompts from public data based on it, providing a stronger baseline.
>
> Q9: Alignment step and prompt embedding latent.
>
>
> A9: **Alignment.** Alignment still contributes to performance. Its gains are smaller in completion tasks (+2.5/+2.2) but larger in instruct tasks (+5.2/+4.6). We further confirm this on MBPP results (4-model average), showing larger gains for instruct than complete (see the figure in this [link](https://anonymous.4open.science/r/PrivCode_Plus-02FE/images/ablation_mbpp.png)).
>
> This is because completion prompts are typically minimal (e.g., short signatures and comments):
>
> ```
> def add(x: int, y: int):
> """Add two numbers x and y
> >>> add(2, 3)
> 5
> """
> ```
>
> Alignment enables PrivCode++ to better leverage rich prompt semantics for higher-quality generation, e.g., in instruct tasks, thus yielding smaller gains when prompts are minimal.
>
>
> **Prompt embedding.** No, using only prompt embedding doesn’t achieve comparable utility. We add a PC-PromptEmb baseline and observe that it performs significantly worse than PrivCode++, since code snippets provide richer syntactic and semantic signals for latent conditioning. Results on Qw2.5-Coder-7B are as follows,
>
> |Method|Instruct||||||Complete||||||
> |-|-|-|-|-|-|-|-|-|-|-|-|-|
> ||HE|HE+|MBPP|MBPP+|BCB-Full|BCB-Hard|HE|HE+|MBPP|MBPP+|BCB-Full|BCB-Hard|
> |PC-PromptEmb|54.9|48.2|65.9|56.3|17.6|2.7|58.5|50.6|73.3|61.1|38.5|7.4|
> |PrivCode++|68.3|60.4|68.0|57.1|21.6|4.7|64.0|56.1|77.8|64.8|43.3|15.5|

---

> > ### Author Rebuttal · Reviewer_tTVa · 2026-04-03
> >
> > I have the following concerns after reading the rebuttal:
> > - From the introduction, I assume that the service provider of the LLM does not want to leak sensitive information from the code in the training set of the LLM. If this is the case, then what part of the code is considered as sensitive information? If it's the whole code snippet, then this is not a real assumption. If it's just a password or PII that is contained inside the snippet, then how do you ensure that that information only exists in one snippet? If that's not the case, the sensitive information is scattered across the dataset (multiple data points may contain similar information), resulting in a breach of the privacy guarantee, since DP-SGD doesn't provide content-DP; it protects the membership of each data point and assumes that each data point is independent.
> > - From the answer to Q2, the data-point level DP only works if there's no correlation between data points. For instance, a codebase may share the same credentials, leading to the same problem of my previous comment.

---

> > > ### Author Response · Authors · 2026-04-04
> > >
> > > We sincerely thank the reviewer for the follow-up. We respond to the concerns below.
> > >
> > > ### **About privacy definition:**
> > >
> > > We agree that the goal is to prevent leakage of sensitive information from the training data, and we propose an approach to address such leakage from both the prompt and the corresponding code.
> > >
> > > **Data characteristics.**
> > > As mentioned in Sec. I, we focus on training models on instruction-following datasets to acquire domain-specific knowledge. These datasets are typically constructed from individual single-turn user–LLM interactions (e.g., CodeAlpaca [a]), where each data sample consists of a natural language instruction and a corresponding solution (i.e., a code snippet). Such data have two properties:
> > > - They are not repository-level and have shorter sequences, making each sample less likely to be truncated into multiple sub-sequences.
> > > - They arise from diverse user tasks and thus exhibit limited correlation across samples [b].
> > >
> > > **Definition of data point.**
> > > We follow prior work [c] and define each protected data point as a truncated (prompt, code) pair, which serves as the atomic protected unit in our DP formulation. Two datasets differ if they vary by one such pair, and all components of our framework are trained under a unified DP-SGD mechanism.
> > >
> > > **Scope of sensitive information.**
> > > We do not assume a prior knowledge of which part of the truncated (prompt, code) pair is sensitive (e.g., specific tokens such as passwords or PII). Instead, we treat the entire data point as a potentially sensitive unit and aim to bound its influence on the trained model. This ensures that any information contained in the pair—regardless of its location—is accounted for under the data point-level DP guarantee, avoiding reliance on identifying specific sensitive tokens and providing uniform protection even when such information is unknown or distributed (e.g., proprietary logic, identifiers, or other sensitive patterns).
> > >
> > >
> > >
> > > ### **About correlation and DP guarantee:**
> > >
> > > We acknowledge that correlations across data points may weaken DP in practice [d], and this issue also arises in tabular [e] and image domains [f]. Prior work typically analyzes the impact of correlations by characterizing different sources of dependency across data points, rather than assuming strict independence.
> > >
> > > As raised in the reviewer’s concern, in instruction-following code generation data, two types of potential correlations may arise.
> > >
> > > - **Within-sample truncation.** A complete sample may be split into multiple DP data points, introducing syntactic and semantic dependencies. However, (prompt, code) pairs are typically shorter and more localized than long-form text or repository-level code, reducing truncation and dependencies. Moreover, such correlations can be mitigated in practice by increasing the maximum sequence length so that each original data sample can be padded into a single DP data point. In our implementation, we choose a moderate value of 1024, resulting in nearly zero truncation in both original and synthetic datasets.
> > >
> > > - **Cross-sample overlap.** Different original data samples (i.e., user-level instruction-following pairs) may share overlapping content.  Prior work [g] shows that while leakage increases in the extreme case of full duplication ($\rho = 1.0$), it remains well below worst-case DP bounds under low correlation (e.g., $|\rho| \le 0.4$), where $\rho$ is the Pearson correlation between data points. Thus, while correlations may weaken the theoretical guarantees, data point-level DP remains effective in providing protection for sensitive information under realistic, limited-correlation settings. This is also supported by our results in Tab. 2: with canaries repeated up to 100 times in the training data, we observe zero leakage, supporting the effectiveness of data point-level DP guarantee under limited repeated patterns.
> > >
> > > **Summary and scope.**
> > > Overall, we follow existing practices of applying DP in the text domain [h], and regard DP as the primary privacy framework that provides worst-case guarantees. Our work therefore focuses on protecting sensitive information at the data point level, rather than addressing all forms of content-level leakage under arbitrary correlations.
> > >
> > > [a] Code Alpaca: An Instruction-following Llama Model for Code Generation
> > >
> > > [b] OpenCodeInstruct: A Large-scale Instruction Tuning Dataset for Code LLMs
> > >
> > > [c] PrivCode: When Code Generation Meets Differential Privacy
> > >
> > > [d] What Does it Mean for a Language Model to Preserve Privacy?
> > >
> > > [e] Correlated Tuple Data Release via Differential Privacy
> > >
> > > [f] Enhancing Correlated Big Data Privacy Using Differential Privacy and Machine Learning
> > >
> > > [g] Impact of Prior Knowledge and Data Correlation on Privacy Leakage: A Unified Analysis
> > >
> > > [h] VaultGemma: A Differentially Private Gemma Model

---

### Official Review · Reviewer_ni2Q · 2026-03-09

**Soundness:** 3
**Presentation:** 3
**Significance:** 2
**Originality:** 2
**Overall Recommendation:** 4
**Confidence:** 3

**Summary:**

This paper studies differentially private code generation in the joint-sensitive setting where both prompts and code snippets are considered sensitive. The paper is well written and clearly structured, making the motivation and methodology easy to follow. The authors propose a two-stage pipeline that introduces latent conditioning to enable code generation without directly using sensitive prompts, and then synthesizes instruction–code pairs for further fine-tuning. The empirical evaluation is relatively extensive, covering multiple benchmarks, models, and ablation studies, as well as privacy leakage analysis. However, the technical novelty appears somewhat limited. The main contribution lies in the overall pipeline design, while several components (e.g., latent conditioning and synthetic data generation pipelines) are relatively common in related work. The privacy analysis also mainly relies on standard DP-SGD guarantees and post-processing properties without introducing new theoretical insights. As a result, while the method shows consistent improvements in the joint-sensitive setting, the gains appear incremental rather than substantial.

**Compliance With Llm Reviewing Policy:**

Affirmed.

**Key Questions For Authors:**

Please address the concerns listed in the Weaknesses section.

**Limitations:**

Yes

**Strengths And Weaknesses:**

S1: Good writing. The paper is clearly structured, and the motivation, method, and experimental settings are presented in an easy-to-follow manner.

S2: Extensive experiments. The authors evaluate the method on multiple benchmarks, models, and settings, and include ablation studies and privacy leakage evaluations.

W1: Technical depth is somewhat limited. The main contribution mainly lies in the overall pipeline design rather than a fundamentally new technical component. Some of the design choices (e.g., latent conditioning and synthetic data generation pipelines) are relatively common in related literature.

W2: The privacy analysis is relatively standard and mainly relies on existing DP-SGD guarantees and post-processing properties. It does not introduce new privacy analysis techniques or theoretical insights.

W3: The empirical improvements appear incremental. While the method consistently outperforms the baselines in the joint-sensitive setting, the margins are generally moderate.

---

> ### Author Rebuttal · Authors · 2026-03-31
>
> Q1: Contribution.
>
> A1: We thank the reviewer for the feedback. We are the first work to explore DP code generation under the joint-sensitive setting, where both prompts and code snippets are treated as private in LLM fine-tuning. Our main contribution lies in the novel pipeline design for code generation, and in leveraging latent conditioning for prompt privacy protection, which has not been explored in prior work. We further validate the effectiveness of PrivCode++ through extensive experiments, demonstrating a strong utility–privacy trade-off.
>
>
> Q2: Privacy analysis.
>
> A2: We follow existing practices of applying DP in the text domain [a] and adopt DP as the standard formal guarantee under worst-case scenarios. For the joint-sensitive setting, we are the first to provide a formal formulation in Sec. 2.2, where we treat each truncated (prompt, code) sequence as a single DP data point, and define neighboring datasets as differing by the presence or absence of one such sequence. And we provide more detailed privacy analysis in App. B.
>
> [a] https://services.google.com/fh/files/blogs/vaultgemma_tech_report.pdf
>
> Q3: Improvements.
>
> A3: Under the joint-sensitive setting, PrivCode++ significantly outperforms DPFT and PC-Uncond, and consistently surpasses the strongest baseline PC-PreEmb, and even outperforms PrivCode on some benchmarks (e.g., on Qwen2.5-Coder-7B, 68.3 vs. 67.1 on HumanEval (instruct) and 64.0 vs. 43.9 on HumanEval (complete)). In terms of privacy, PrivCode++ consistently achieves 0 leakage, while other baselines exhibit leakage under different scenarios.

---

### Official Review · Reviewer_LwGC · 2026-03-12

**Soundness:** 2
**Presentation:** 3
**Significance:** 2
**Originality:** 2
**Overall Recommendation:** 4
**Confidence:** 3

**Summary:**

This paper proposes a novel framework for effective privacy-preserving LLM coding fine-tuning, named PrivCode++. The authors treat both the prompt and the generated code snippets as private information and introduce the Privacy-Free Latent Conditioning (PrivLC) module to obtain latent conditional embeddings, which can effectively guide the generation of synthetic coding snippets for downstream LLM fine-tuning. Moreover, they utilize a prompt extractor to generate corresponding prompts for each synthetic code snippet, further enhancing the fine-tuning process of the LLM. Extensive experimental results demonstrate the proposed approach achieve better fine-tuning performance without privacy leakage.

**Compliance With Llm Reviewing Policy:**

Affirmed.

**Final Justification:**

The author response addresses some of my concerns. However, some other issues still remain. Firstly, the evaluation should be extended to more domain-specific tasks beyond the coding domain, because coding is not inherently privacy-sensitive. Secondly, the current version lacks analysis of several important hyperparameters. Therefore, I maintain my original score of weak accept.

**Key Questions For Authors:**

1.Does Figure 4 indicate that the PrivCode baseline can generate synthetic data that is more similar to the original private data than PrivCode++?

2.There is a typo in Line 719 for the reference of The Stack.

**Limitations:**

Yes.

**Strengths And Weaknesses:**

Strengths:

1.The proposed challenge for existing data-synthesis-based methods for privacy-preserving LLM fine-tuning is interesting and realistic. Indeed, in real-world applications, both the prompt and the response should be considered private.

2.The idea of sampling random conditional latent vectors and decoding them into prefixes for data synthesis is interesting, and it can address the problem of generating synthetic data when prompts are not provided.

3.The paper is well organized and easy to follow.

Weaknesses:

1.Some experiment settings are not convincing. For example, the authors use a strong expert LLM, Llama-3.1-70B, to extract synthetic prompts. Given that such a powerful LLM is already employed during the data synthesis stage, it raises the question of why not directly use the expert LLM to unconditionally generate synthetic data for fine-tuning. Since coding data is typically not highly privacy-sensitive and expert LLMs are already well pre-trained for coding tasks, they could potentially generate a large amount of high-quality synthetic data. This could serve as an important baseline for comparison in the paper.

2.Indeed, the focus domain of this paper is coding, which is generally less privacy-sensitive compared to domains such as healthcare or finance. Therefore, those highly privacy sensitive domain should be evaluated to demonstrate your contribution for privacy protection in real life application.

3.Some important experimental settings are missing. In Section 4.3, the authors inject several canary samples containing private information into the OSS-Instruct PII Dataset. However, the paper does not specify how many canary samples are injected or what proportion they occupy in the overall OSS-Instruct PII Dataset. Moreover, the paper does not provide sufficient details about the evaluation metric, the leakage rate, making it unclear how it is computed.

4.The proposed method contain many hyper-parameters which should be further analyzed, such as the length of the conditional prefix e_{prefix}, the total amount of the synthetic data and so on.

---

> ### Author Rebuttal · Authors · 2026-03-31
>
> Q1: Expert LLM.
>
> A1: We thank the reviewer for the suggestion. As noted, expert LLMs are indeed well pre-trained for general coding tasks; however, when specific domain knowledge is required, they typically lack the ability to directly generate high-fidelity data aligned with the target domain, and thus cannot reliably produce synthetic data that matches the distribution of the specific-domain dataset. To validate this, we include an additional baseline **PC-Expert**, where we use Llama-3.1-70B-Instruct to unconditionally generate synthetic data (only given the minimal coding prefix, i.e., "\```python\n", "```cpp\n", sampled according to the language distribution in the original dataset), while keeping the remaining pipeline identical to PrivCode++; the results on Qw2.5-Coder-7B are as follows, showing **inferior** performance compared to our method.
>
> |Method|Instruct||||||Complete||||||
> |---------------|--------------|--------|--------|--------|--------|--------|--------------|--------|--------|--------|--------|--------|
> ||HumanEval||MBPP||BigCodeBench||HumanEval||MBPP||BigCodeBench||
> ||HE|HE+|MBPP|MBPP+|Full|Hard|HE|HE+|MBPP|MBPP+|Full|Hard|
> |PC-Expert|58.5|54.9|64.6|51.9|18.2|3.4|31.1|32.3|25.9|32.0|14.0|4.7|
> |PrivCode++|68.3|60.4|68.0|57.1|21.6|4.7|64.0|56.1|77.8|64.8|43.3|15.5|
>
> Moreover, the external LLM is a **replaceable utility module**; we use Llama-3.1-70B-Instruct primarily to demonstrate the upper-bound performance. We conduct additional experiments, replacing it with a smaller model (e.g., Llama-3.1-8B-Instruct) in either the prompt extraction or round-trip filtering module, which yield **comparable results**, as shown below:
>
> | External LLM | Instruct-Humaneval | Instruct-Humaneval++ | Complete-Humaneval | Complete-Humaneval++ | Instruct-MBPP | Instruct-MBPP++ | Complete-MBPP | Complete-MBPP++ |
> | ----------------------------------------- | ------------------ | -------------------- | ------------------ | -------------------- | ------------- | --------------- | ------------- | --------------- |
> | Llama-3.1-70B-Instruct | 68.3 | 60.4 | 68.0 | 57.1 | 64.0 | 56.1 | 77.8 | 64.8 |
> | Llama-3.1-8B-Instruct (Prompt Extractor)  | 67.1 | 59.8 | 66.4 | 56.1 | 64.0|55.5 | 77.0 | 63.5 |
> | Llama-3.1-8B-Instruct (Round-trip Filter) | 64.0 | 55.5 | 66.7 | 55.0 | 61.0 | 54.9 | 73.8 | 60.8 |
>
> This is because the quality of synthetic data is mainly determined by code generation under latent conditioning, while summarizing code snippets into instruction prompts is relatively simple and can be effectively handled by smaller models, making reliance on large external LLMs less critical.
>
> Q2: Sensitive domain.
>
> A2: Many types of real-world domain-specific knowledge rely on learning from code-form data [a], and our method provides a general privacy-preserving framework for code, which can be applied to sensitive domains such as healthcare and finance.
>
> [a] https://arxiv.org/pdf/2504.04030
>
> Q3: Canary details.
>
> A3: We clarify that the number of injected canaries is controlled via the repetition count {5, 10, 100}, as described in Sec. 4.3, where each canary sample is inserted multiple times into the training set. As described in App. C.3, the leakage rate is computed using a category-level metric, defined as the percentage of canary categories whose tokens appear in any generated output, following prior work [b]. We further note in App. F that each canary category contains a single unique canary sample; a category is considered leaked if its canary token is exactly reproduced in the generated output.
>
> We agree that the presentation could be clearer regarding the absolute number of canary samples and their proportion in the dataset. We include these details as follows,
>
> |Repetition|Absolute Count|Proportion|
> |------------|----------------|------------|
> |1|5|0.99%|
> |10|50|9.09%|
> |100|500|50%|
>
> [b] https://aclanthology.org/2023.acl-long.74.pdf
>
> Q4: More detailed hyper-parameters analysis.
>
> A4: We will include more detailed hyper-parameter analysis in the revision, including factors such as the length of the conditional prefix $e_{\text{prefix}}$ and the total amount of synthetic data.
>
> Q5: Fidelity of PrivCode++ compared with PrivCode.
>
> A5: Yes. This is expected since PrivCode operates under a relaxed setting where prompts are public, allowing the model to directly condition on original prompt information and thus better match the original data distribution. In contrast, PrivCode++ works under a stricter joint-sensitive setting without direct access to prompts. Despite this, it still achieves competitive fidelity while providing stronger privacy guarantees.
>
> Q6: Typo.
>
> A6: We thank the reviewer for pointing this out. We will correct the typo in the reference to The Stack in the revised version.

---

> > ### Author Rebuttal · Reviewer_LwGC · 2026-04-01
> >
> > Thanks for your response. I keep my original score.

---

### Official Review · Reviewer_jVsH · 2026-03-23

**Soundness:** 3
**Presentation:** 3
**Significance:** 3
**Originality:** 3
**Overall Recommendation:** 4
**Confidence:** 3

**Summary:**

The authors propose PrivCode++, a two-stage framework: a privacy-sanitizing stage and a utility-boosting stage to handle joint-sensitive code generation, where both prompts and code are sensitive.

**Compliance With Llm Reviewing Policy:**

Affirmed.

**Final Justification:**

My concerns have been adequately addressed.

**Key Questions For Authors:**

1. The canary analysis is helpful, could the authors discuss broader attacks beyond canary leakage?
2. The paper mainly reports results at ε = 4; how does the method perform under stricter privacy budgets?
3. How much of the performance gain actually comes from the latent conditioning itself, versus relying on public LLM for instruction extraction and filtering?

**Limitations:**

Please check weaknesses

**Strengths And Weaknesses:**

Strengths:
- The motivation is clear that prompts can be sensitive and require protection
- The two-stage framework is easy to follow and clear
- Good evaluation results compared to baselines

Weaknesses:
- The privacy evaluation is limited to canary-based memorization tests; incorporating broader evaluations such as membership inference attacks would strengthen the analysis
- Results focus mainly on ε = 4, a moderate privacy setting. A clearer characterization of the privacy–utility trade-off is needed to assess performance under stronger privacy constraints
- The approach relies heavily on external LLMs and a multi-stage filtering pipeline, which may limit practicality and make reproduction or deployment costly

---

> ### Author Rebuttal · Authors · 2026-03-31
>
> Q1: Broader attacks.
>
> A1: We thank the reviewer for the suggestion. We further include a classical loss-based membership inference attack (MIA). Specifically, we perform the attack on the premium model trained on the canary-injected dataset. We first partition the original sensitive data into member and non-member sets based on whether each sample appears in the training set. For the member portion, we additionally inject canaries to align with the actual training distribution. We then use per-sample loss as the attack score, and report TPR@1%FPR and TPR@10%FPR as follows:
>
> |Method|Qwen2.5-Coder-7B||DS-Coder-6.7B||
> |-|-|-|-|-|
> ||TPR@10%FPR|TPR@1%FPR|TPR@10%FPR|TPR@1%FPR|
> |NonDPFT|69.9%|49.0%|59.8%|59.1%|
> |PrivCode|12.5%|0.0%|13.1%|0.0%|
> |PrivCode++|2.6%|0.0%|3.7%|0.0%|
>
> The relatively low TPR@10%FPR (below 10%) is due to the inherently higher loss of the canary samples themselves. This is because the canaries are generated in a few-shot manner using the original sensitive data as examples, and thus follow a distribution that is slightly shifted from the original sensitive data. Overall, the results further support the effectiveness of our privacy protection.
>
> Q2: Privacy budget analysis.
>
> A2: Thanks for the suggestion. The utility under different privacy budgets and the corresponding analysis are already provided in Fig. 3 and App. E.4. We additionally report the privacy protection performance under other $\epsilon$ values as follows. Under canary repetition count as 100, when $\epsilon=\{1,4\}$, PrivCode++ provides strong privacy protection; when $\epsilon=\infty$, privacy leakage can occur. Therefore, we recommend using a moderate privacy budget (e.g., $\epsilon=4$) in practice, following prior works (DP text synthesis [a], DP code generation [b]).
>
> |Canary|Q2.5 $\epsilon=1$|Q2.5 $\epsilon=10$|Q2.5 $\epsilon=\infty$|Gem $\epsilon=1$|Gem $\epsilon=10$|Gem $\epsilon=\infty$|CQ1.5 $\epsilon=1$|CQ1.5 $\epsilon=10$|CQ1.5 $\epsilon=\infty$|DS $\epsilon=1$|DS $\epsilon=10$|DS $\epsilon=\infty$|
> |:---|:---:|:---:|:---:|:---:|:---:|:---:|:---:|:---:|:---:|:---:|:---:|:---:|
> |**Joint**|0\%|0\%|40\%|0\%|0\%|0\%|0\%|0\%|20\%|0\%|0\%|20\%|
> |**Prompt**|0\%|0\%|0\%|0\%|0\%|20\%|0\%|0\%|0\%|0\%|0\%|0\%|
> |**Code**|0\%|0\%|20\%|0\%|0\%|0\%|0\%|0\%|0\%|0\%|0\%|20\%|
>
> [a] https://aclanthology.org/2023.acl-long.74.pdf
>
> [b] https://arxiv.org/pdf/2512.05459
>
>
>
> Q3: External LLMs.
>
> A3: We acknowledge that using external LLMs introduces additional cost. However, they are not the primary source of performance gain. In our experiments, other baselines also use the same Llama-3.1-70B-Instruct as a controlled factor (e.g., PC-Uncond and PC-PreEmb for prompt extraction, and PrivCode, PC-Uncond, and PC-PreEmb all use it for round-trip filtering), indicating that the performance gap (improving Pass@1 score by up to 8.2 on instruction-following tasks and 19.3 on code completion tasks) mainly comes from our pipeline design and the Privacy-Free Latent Conditioning (PrivLC) module, rather than the external LLM itself.
>
> Moreover, this component is fully replaceable; in this paper, we use the 70B model primarily to demonstrate the upper-bound performance. We further replace Llama-3.1-70B-Instruct with a smaller model, such as Llama-3.1-8B-Instruct, by substituting the external LLM in either the prompt extraction or the round-trip filtering module separately for ablation, and observe comparable results.
>
> For prompt extraction, we fine-tune Qwen2.5-Coder-7B, and evaluate on the EvalPlus benchmark as follows,
>
> | External LLM | Instruct-Humaneval | Instruct-Humaneval++ | Complete-Humaneval | Complete-Humaneval++ | Instruct-MBPP | Instruct-MBPP++ | Complete-MBPP | Complete-MBPP++ |
> |--------------|-------------------|---------------------|--------------------|----------------------|---------------|-----------------|----------------|------------------|
> | Llama-3.1-70B-Instruct | 68.3  | 60.4 |  68.0 |  57.1 | 64.0| 56.1| 77.8| 64.8 |
> | Llama-3.1-8B-Instruct | 67.1 | 59.8 | 66.4 | 56.1 | 64.0|55.5 | 77.0 | 63.5 |
>
> as same, for round-trip validation,
>
> |External LLM|Instruct-Humaneval|Instruct-Humaneval++|Complete-Humaneval|Complete-Humaneval++|Instruct-MBPP|Instruct-MBPP++|Complete-MBPP|Complete-MBPP++|
> |--------------|-------------------|---------------------|--------------------|----------------------|---------------|-----------------|----------------|------------------|
> |Llama-3.1-70B-Instruct|68.3|60.4|68.0|57.1|64.0|56.1|77.8|64.8|
> |Llama-3.1-8B-Instruct|64.0|55.5|66.7|55.0|61.0|54.9|73.8|60.8|
>
>
> This is because the quality of synthetic data is mainly determined by the code generation under latent conditioning, while the task of summarizing code snippets into instruction prompts in both steps is relatively simple and can be effectively handled by smaller models, making the reliance on strong external LLMs less critical.

---

> > ### Author Rebuttal · Reviewer_jVsH · 2026-04-02
> >
> > Thanks for your response. I will update the score.

---

### Decision · Program_Chairs · 2026-04-30

**Decision:**

Accept (regular)

**Comment:**

This paper received a divergent set of scores (2, 4, 4, 4). While it might not be directly visible to authors,  Reviewer ni2Q acknowledged the rebuttal. Note that ICML this year restricted the discussion between reviewers and authors to one additional round for a good reason. There was a concern on citation: Liu et al. (2024), which was cited as published in NeurIPS 2024, was actually published in NeurIPS 2023. However, AC assumed good intention and the citation is an honest mistake that should be corrected, instead of rejecting based on this concern.

Reviewers raised concerns on code sensitivity, differential privacy protection, hyperparameters, which are partially solved during rebuttal. Reviewers acknowledged that protecting private prompts (compared to PrivCode assuming public prompts) are valuable. Two-stage DP generators are also explored for general DP synthetic data generation not for code, and I would encourage the authors to discuss these related works, see for example [how to dp-fy your data](https://arxiv.org/abs/2512.03238) Section 5 & 6. Overall, the paper is timely and valuable, while the contributions are incremental. This is a borderline paper.